# Remodelling of tumour microenvironment by microwave ablation potentiates immunotherapy of AXL-specific CAR T cells against non-small cell lung cancer

Bihui Cao [1,2,11], Manting Liu[1,2,11], Lu Wang[2], Kangshun Zhu[1], Mingyue Cai[1], Xiaopei Chen[2], Yunfei Feng[2], Shuo Yang[3,4], Shengyu Fu[3], Cheng Zhi[5], Xiaodie Ye[2], Jian Zhang[2], Zhiru Zhang[2], Xin Yang[6], Ming Zhao[7], Qingde Wu[8], Linfeng Xu[9], Lili Yang[10], Hui Lian[1], Qi Zhao [3] ✉ & Zhenfeng Zhang [1,2] ✉

The complex immunosuppressive tumour microenvironment (TME) and lack of tumour-specific targets hinder the application of chimeric antigen receptor (CAR) T cells in the treatment of solid tumours. Combining local treatment with CAR T cell immunotherapy may regulate the TME and enhance the killing potency of CAR T cells in solid tumours. Here, we show that AXL, which is highly expressed in non-small cell lung cancer (NSCLC) but not in normal tissues, might be a target for CAR T cell therapy. AXL-CAR T cells alone cause moderate tumour regression in subcutaneous and pulmonary metastatic lung cancer cell-derived xenograft models. Combination of microwave ablation (MWA) and AXL-CAR T cells have superior antitumour efficacy. MWA enhances the activation, infiltration, persistence and tumour suppressive properties of AXL-CAR T cells in AXL-positive NSCLC patient-derived xenograft tumours via TME remodelling. The combination therapy increases the mitochondrial oxidative metabolism of tumour-infiltrating CAR T cells. Combination treatment induces significant tumour suppression without observed toxicities in humanized immunocompetent mice. The synergistic therapeutic effect of MWA and AXL-CAR T cells may be valuable for NSCLC treatment.

The chimeric antigen receptor (CAR) T cell therapy is a promising approach for tumour treatment, as exemplified by the successful application of CD19-CAR T cells in B cell derived haematological malignancies[1,2]. Extending the clinical application of CD19-CAR T cell therapy to solid tumours has been actively investigated[3]. However, the results of solid tumour immunotherapy with CAR T cells are not as notable as the outcomes for haematological cancers[4]. The lack of tumour-specific antigen, inefficient infiltration of CAR T cells, potential

[1]Department of Minimally Invasive Interventional Radiology, the Second Affiliated Hospital of Guangzhou Medical University, Guangzhou 510260, China. [2]Central Laboratory, the Second Affiliated Hospital of Guangzhou Medical University, Guangzhou 510260, China. [3]Institute of Translational Medicine, Cancer Centre, Faculty of Health Sciences, University of Macau, Macau SAR 999078, China. [4]Faculty of Pharmacy, Guangzhou Medical University, Guangzhou 511436, China. [5]Department of Pathology, the Second Affiliated Hospital of Guangzhou Medical University, Guangzhou 510260, China. [6]Department of Thoracic Surgery, the Second Affiliated Hospital of Guangzhou Medical University, Guangzhou 510260, China. [7]Department of Interventional Radiology, Cancer Center, Sun Yat-sen University, Guangzhou 510060, China. [8]Department of Radiology, Shunde Chinese Medicine Hospital, the Affiliated Hospital of Traditional Chinese Medicine University of Guangzhou, Foshan 528000, China. [9]Department of Interventional Therapy, Sun Yat-sen Memorial Hospital, Sun Yat-sen University, Guangzhou 510120, China. [10]Department of Nutrition, School of Public Health, Sun Yat-sen University, Guangzhou 510080, China. [11]These authors contributed equally: Bihui Cao, Manting Liu. ✉e-mail: qizhao@um.edu.mo; zhangzhf@gzhmu.edu.cn

side effects, antigen heterogeneity, and immunosuppressive tumour microenvironment (TME) are currently considered as the main obstacles for CAR T therapy in solid tumours[5]. Among these roadblocks, the complex and heterogeneous TME of solid tumours plays an essential role in tumour initiation, progression, and resistance to therapeutics. Intrinsic TME is largely composed of extracellular matrix and immunosuppressive modulators and characterised by aberrant cell proliferation and vascularisation, leading to hypoxia and altered nutrient availability, which impedes the infiltration, survival, and effective function of CAR T cells in solid tumours[6]. In addition to screening novel tumour-specific antigens and optimising intrinsic structure and function of CAR T cells, remodelling of TME in solid tumours to improve CAR T cell potency has been proposed[7]. A recent preclinical finding suggested that mild heating of tumour by local photothermal therapy triggered physicochemical and physiological changes that increased infiltration and accumulation of CSPG4-specific CAR T cells in melanoma[8]. Moreover, cytokine-armed oncolytic adenoviruses facilitated the accumulation and effective function of CAR T cells within pancreatic cancer tissue[9]. These results suggested that combination of CAR T immunotherapy and local treatment could be a particularly promising strategy for solid tumours.

In clinical practice, local ablative therapies, such as microwave ablation (MWA), have been widely applied to treat various benign and malignant tumours of the lung, liver, kidney, bone and soft tissues[10]. MWA uses electromagnetic waves to generate heat and also kills cells by direct and indirect mechanisms, causing rapid hyperthermic lethal or sublethal injury to ablated cells, which affects the TME and damages cells at the membrane and subcellular levels. Unfortunately, local tumour residue and marginal recurrence are common complications caused by incomplete ablation, which affects the therapeutic efficacy and prognosis in the clinical setting[11]. Recently, an increasing amount of evidence has indicated that various immunomodulatory factors, such as tumour antigens, danger signals and cytokines, are released after MWA and play a key role in stimulating the antitumour immune response[12]. However, the mechanism underlying the immune boosting effect is not entirely understood, and the immune response produced by ablation alone is often too modest to completely expunge established tumours[13]. Several preclinical and clinical studies have suggested that MWA can induce therapeutically effective systemic antitumour immune response if combined with appropriate immunomodulators[12]. Consequently, pairing MWA with CAR T cell immunotherapy might have a synergistic therapeutic value and is worth further exploration with regards to its efficacy in the treatment of solid tumours.

Our previous study demonstrated that the activation of AXL was required for the epithelial-mesenchymal transition and promoted erlotinib resistance in non-small cell lung cancer (NSCLC) with mutated epidermal growth factor receptor (EGFR)[14]. It was also reported that AXL is a typical oncogene, which is overexpressed in many human cancers[15,16], including lung, breast, colon, prostate, gastric, pancreatic, renal, oesophageal, thyroid, liver, and ovarian cancers. And high level of AXL expression is associated with poor prognosis in different types of cancers[15]. Because of AXL's discriminative expression and its implication in both tumour biology and chemotherapeutic resistance, a therapeutic that targets AXL could be valuable cancer therapy[17]. Although recent studies demonstrated that AXL may also play an important role in noncancer cells, including neurons, endothelial cells, and immune cells, with roles in homeostasis, angiogenesis, neurogenesis, and innate immunity that may raise concern for potential haematologic and/or immune side effects[17–19], a series of AXL-targeted small-molecule inhibitors have been investigated in preclinical and clinical trials, including R428 and BPI-9016M, which showed favourable safety and pharmacokinetic profiles in NSCLC patients[20,21]. Moreover, a high number of AXL-targeted monoclonal antibodies, such as hMAb173 for renal cell carcinoma[22], D9 and E8 for pancreatic cancer[23], and YW327.6S2 for NSCLC and breast cancer[24], have shown anticancer efficacy in preclinical and earlier clinical trial stages[16,17]. In addition, AXL-targeted CAR T cells showed killing effects in breast cancer models in vivo and various tumour models in vitro[25–27]. However, those monotherapies exhibited only moderate antitumour efficacy in previous studies, and none of the studies focused on the use of AXL-targeting CAR T cells for the treatment of NSCLC[16,28]. There is a clear unmet medical need to improve the response of AXL-CAR T cells in NSCLC. In light of the potential immune-boosting capability of MWA and the hypofunction of CAR T cells in solid tumours, the combination of MWA and AXL-CAR T cells may be an attractive approach to treat NSCLC.

In this work, we demonstrate that AXL might be an ideal target for CAR T cell immunotherapy in NSCLC. We design AXL-directed CAR-T cells and show that their anti-tumour activity can be improved in combination with MWA in several preclinical lung cancer models.

## Results

### AXL expression profiles in normal and NSCLC human tissues, and in cell lines

Immunohistochemistry (IHC) staining showed that weak (1+) AXL expression was observed in only two colorectal tissues among 208 human samples from 22 types of normal organs or tissues (Supplementary Fig. 1 and Supplementary Table 1). AXL protein was detected by IHC in 62 out of 90 NSCLC samples (69%). In 30 out of the 62 positive cases (48%), AXL protein expression was moderate (2+) to strong (3+), whereas it was weak (1+) in the 32 remaining cases (Fig. 1a and Supplementary Table 2). Notably, more cases with higher AXL expression were found by IHC in EGFR tyrosine kinase inhibitor (EGFR TKI)-resistant lung cancer tissues (Supplementary Table 2). Moreover, western blotting (WB) and IHC showed that the AXL expression level in EGFR TKI-resistant NSCLC tissues was significantly higher than that in adjacent normal tissues (Fig. 1b, c and Supplementary Fig. 2 and Supplementary Table 3). Among the available lung cancer cell lines, AXL was detected at high levels by WB and flow cytometry in A549 and HCC827-ER3 cells (erlotinib-resistant cell line, according to previous reports) but not in HCC827 cells (Fig. 1d, e). These results demonstrated that AXL was highly expressed in NSCLC tissues and cell lines, but rarely in normal tissues, indicating that AXL might be an ideal target for CAR T immunotherapy against lung cancer.

### AXL-CAR T cells induce moderate tumour regression in subcutaneous and pulmonary metastatic lung cancer xenograft models

We first verified the affinity of anti-AXL single-chain fragment variables (scFvs) (Supplementary Fig. 3), generated three kinds of AXL-specific CAR T cells (Supplementary Fig. 4a–d) and examined their antitumour cytotoxicity and cytokine secretion in vitro against NSCLC cell lines (Supplementary Fig. 4e–g). We found that all three kinds of AXL-CAR T cells had obvious intrinsic target-dependent cytotoxicity and secreted cytokines when co-cultured with AXL-positive NSCLC cell lines. The YW327.6S2-CAR T cells showed the strongest killing potency among the three AXL-CAR T cells (Supplementary Fig. 4f, g) and were selected for further studies.

Then we tested the antitumour efficacy of AXL-CAR T cells in vivo. AXL- or CD19-CAR T cells were used to treat mice bearing established subcutaneous tumour xenografts formed by A549 or HCC827-ER3 cells (Fig. 2a, b). The calculated tumour volume grew slowly in the AXL-CAR T group, but increased dramatically compared to the baseline (50 mm³) in the mock (phosphate buffered saline, PBS) and CD19-CAR T groups (Fig. 2c, d). Correspondingly, the measured average tumour weights in the AXL-CAR T group (A549 0.28 g, HCC827-ER3 0.40 g), were significantly lower than those in the CD19-CAR T (A549 0.68 g, HCC827-ER3 0.82 g) and mock (A549 0.72 g, HCC827-ER3 1.00 g) groups (Fig. 2e, f).

To determine whether AXL-CAR T cells can inhibit distant metastasis of NSCLC, we developed an A549 GL-bearing pulmonary

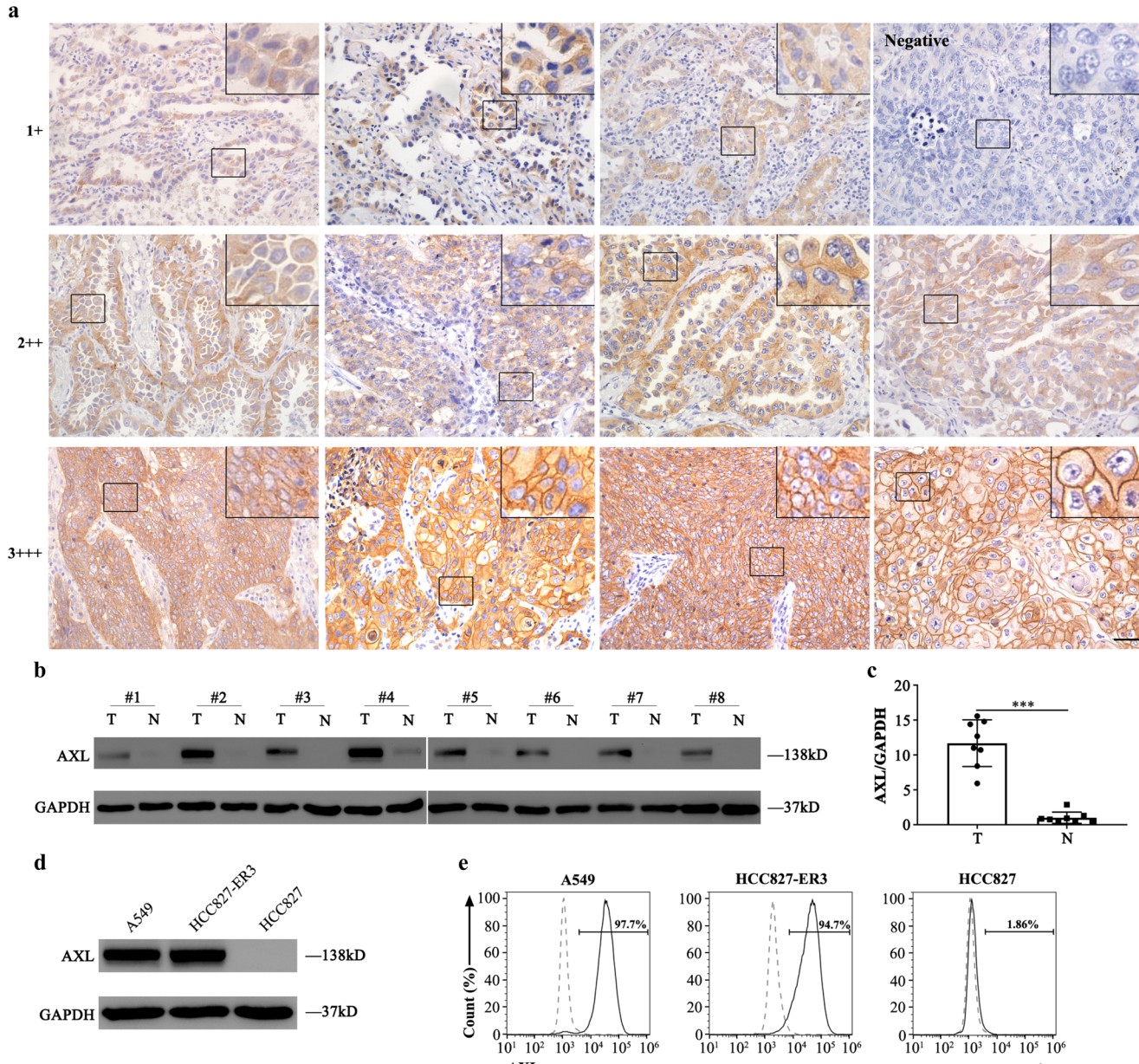

**Fig. 1 | AXL expression in human normal and NSCLC tissues and lung cancer cell lines. a** Detection of AXL expression in NSCLC samples by IHC. Representative images (magnification × 400) are shown (*n* = 90). Scale bar represents 100 μm. **b, c** Detection and quantification of AXL expression by western blotting in eight pairs of EGFR-TKI resistant NSCLC tissues (T) and adjacent normal lung tissues (N). Each group *n* = 8. One of three repetitions with similar results is shown and presented as the mean ± SD (T vs N *p* < 0.0001, two-sided unpaired *t*-test). **d, e** Detection of AXL expression by western blotting and flow cytometry in human NSCLC cell lines A549, HCC827-ER3, and HCC827. One of three repetitions with similar results is shown here. *\*p* < 0.05, *\*\*p* < 0.01, *\*\*\*p* < 0.001. Source data are provided as a Source Data file.

metastatic model (Fig. 2g). Fourteen days after tumour inoculation via the tail vein, tumour cells were detected in the lungs (Fig. 2h), mimicking the pulmonary metastasis of NSCLC. We infused $1 \times 10^7$ AXL- or CD19-CAR T cells intravenously into the pulmonary tumour-bearing mice on the same day. Bioluminescence imaging (BLI) demonstrated that AXL-CAR T cells partially suppressed the pulmonary tumour in the majority of mice on day 38, whereas CD19-CAR T or PBS could not control tumour progression (Fig. 2h, i). Notably, tumour cells that colonised the lungs caused mouse death within 62 days in the mock and CD19-CAR T groups. In contrast, mice in the AXL-CAR T group continued to survive until day 90 (Fig. 2j).

Collectively, our data demonstrated that AXL-CAR T cells alone had a moderately inhibitory, but not a completely suppressive effect against AXL-positive NSCLC subcutaneous and pulmonary metastatic tumours in vivo. Thus, we proceeded to explore whether

combining MWA with AXL-CAR T cells could enhance the antitumour activity against NSCLC.

## Combination of MWA and AXL-CAR T cells exhibits superior and safe local and systemic antitumour activity in NSCLC CDX tumour models

To evaluate the therapeutic effect of AXL-CAR T cells combined with MWA, subcutaneous tumour models were first constructed by injecting HCC827-ER3, A549, or HCC827 cells. The following related study was initiated when the tumour volume reached about 200 mm³. According to the power-time gradient ablation research, we selected 10 W and 40 s as parameters sufficient to achieve partial ablation, which faithfully mimicked the tumour residue after MWA therapy in clinical situations (Supplementary Fig. 5). The inoculated NSG mice were then randomly allocated to the following groups: mock (PBS),

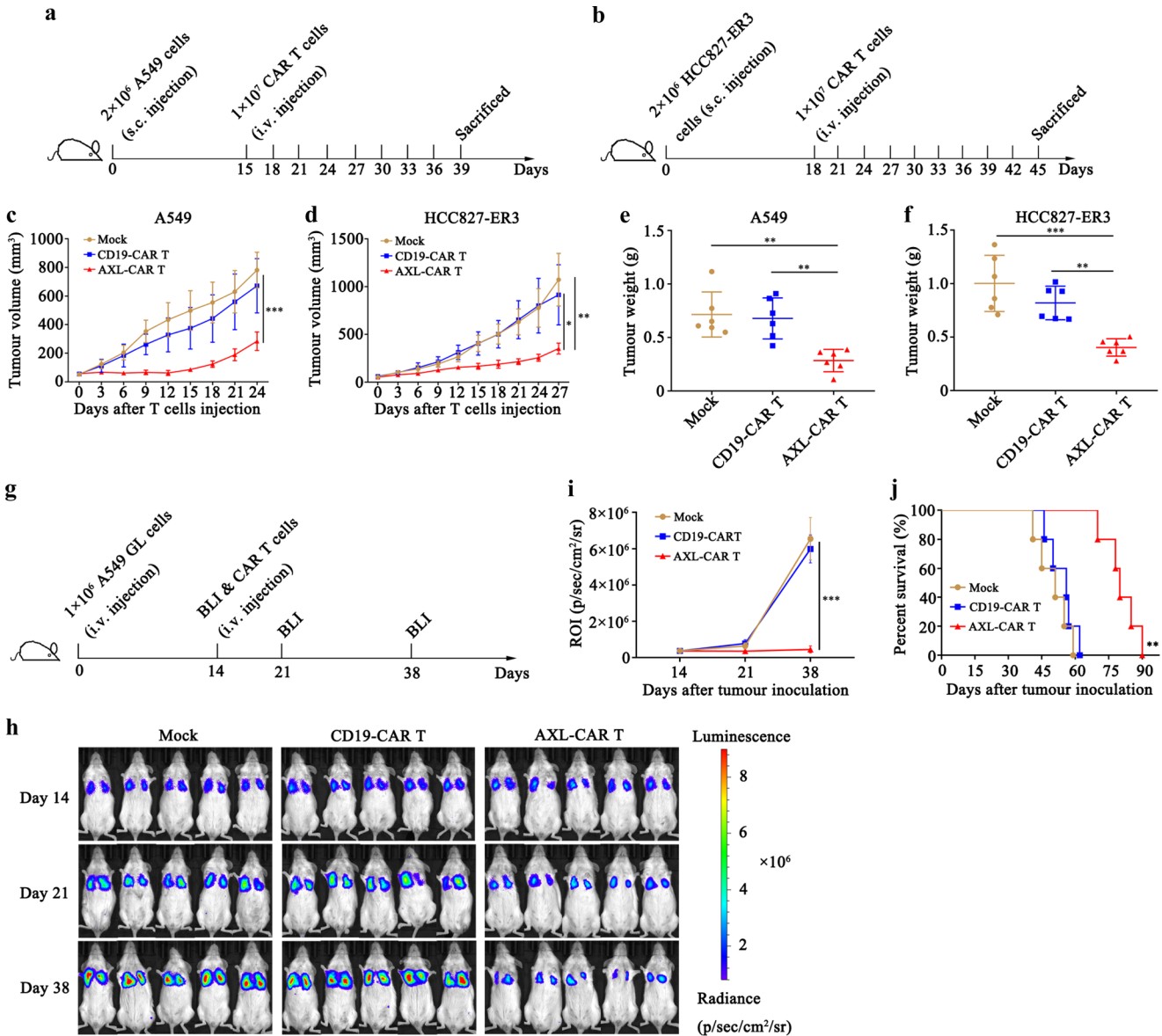

**Fig. 2 | AXL-CAR T cells show moderate antitumour activity in subcutaneous and pulmonary metastasis mouse models. a, b** Schemes of experiments with tumour-bearing mice. NSG mice received a subcutaneous injection of $2 \times 10^6$ A549 or HCC827-ER3 cells. When tumour volume reached about 50 mm³, $1 \times 10^7$ AXL-CAR T cells or CD19-CAR T cells were injected through the tail vein and tumour volume was measured every three days ($n = 6$ mice per group). **c, d** Tumour volume in mice injected subcutaneously with A549 or HCC827-ER3 cells. (**c:** Mock/CD19-CAR T vs AXL-CAR T $p < 0.0001$. **d:** Mock vs AXL-CAR T $p = 0.003$, CD19-CAR T vs AXL-CAR T $p = 0.02$, two-way ANOVA with Tukey's multiple comparisons test). **e, f** Tumour weight in mice injected subcutaneously with A549 or HCC827-ER3 cells at the end point. (**e:** Mock vs AXL-CAR T $p = 0.002$, CD19-CAR T vs AXL-CAR T $p = 0.004$. **f:** Mock vs AXL-CAR T $p < 0.0001$, CD19-CAR T vs AXL-CAR T $p = 0.004$, one-way ANNOVA with Tukey's multiple comparisons test). **g** Schematic representation of the pulmonary metastasis experiment. **h** Bioluminescence imaging (BLI) of mice intravenously injected with A549 GL cells and treated with CD19- or AXL-CAR T cells. Briefly, NSG mice ($n = 5$ mice per group) received an intravenous injection of $1 \times 10^6$ A549 GL cells. After 14 days, $1 \times 10^7$ AXL-CAR T cells or CD19-CAR T cells were injected through the tail vein, and BLI was performed on days 14, 21, and 38. **i** Statistical analysis of the regions of interest at each time point. (Mock/CD19-CAR T vs AXL-CAR T $p < 0.0001$, two-way ANOVA with Tukey's multiple comparisons test). **j** Survival curve of mice with pulmonary metastasis. Statistical significance of the difference was analysed using the log-rank (Mantel-Cox) test ($p = 0.002$). Data are presented as mean ± SD of indicated samples (**c–f, i**). *$p < 0.05$, **$p < 0.01$, ***$p < 0.001$. Source data are provided as a Source Data file.

CD19-CAR T cells ($10^7$ cells, i.v.), AXL-CAR T cells ($10^7$ cells, i.v.), MWA (10 W, 40 s), MWA (10 W, 40 s) combined with CD19-CAR T cells ($10^7$ cells, i.v.), MWA (10 W, 40 s) combined with AXL-CAR T cells ($10^7$ cells, i.v.) (Fig. 3a). On day 54, the mean weights of the harvested tumours at the end point were 0.02 g (HCC827-ER3) and 0.05 g (A549) in the combination group, which was significantly smaller than other groups (Fig. 3b–e). However, both MWA and combination therapy failed to suppress HCC827 (AXL negative) established tumour (Supplementary Fig. 6). In addition, AXL-CAR T cells prevented distant metastasis in HCC827-ER3 models, whereas metastatic nodules in the lung were

observed in the mock, CD19-CAR T, MWA combined with CD19-CAR T and MWA groups (Fig. 3f, g). IHC results verified that the metastatic tumours were HLA-A positive and therefore originated from human HCC827-ER3 cells (Fig. 3f). Correspondingly, the weights of the harvested lungs were significantly heavier in the mock (0.36 g) and CD19-CAR T (0.37 g) groups than that in other groups (Fig. 3h). Importantly, expression levels of Ki-67, a marker of proliferating cells, were clearly lower in the combination and AXL-CAR T groups than those in other groups (Supplementary Fig. 7). These findings indicated that AXL-CAR T cells successfully inhibited tumour growth. Furthermore, no obvious

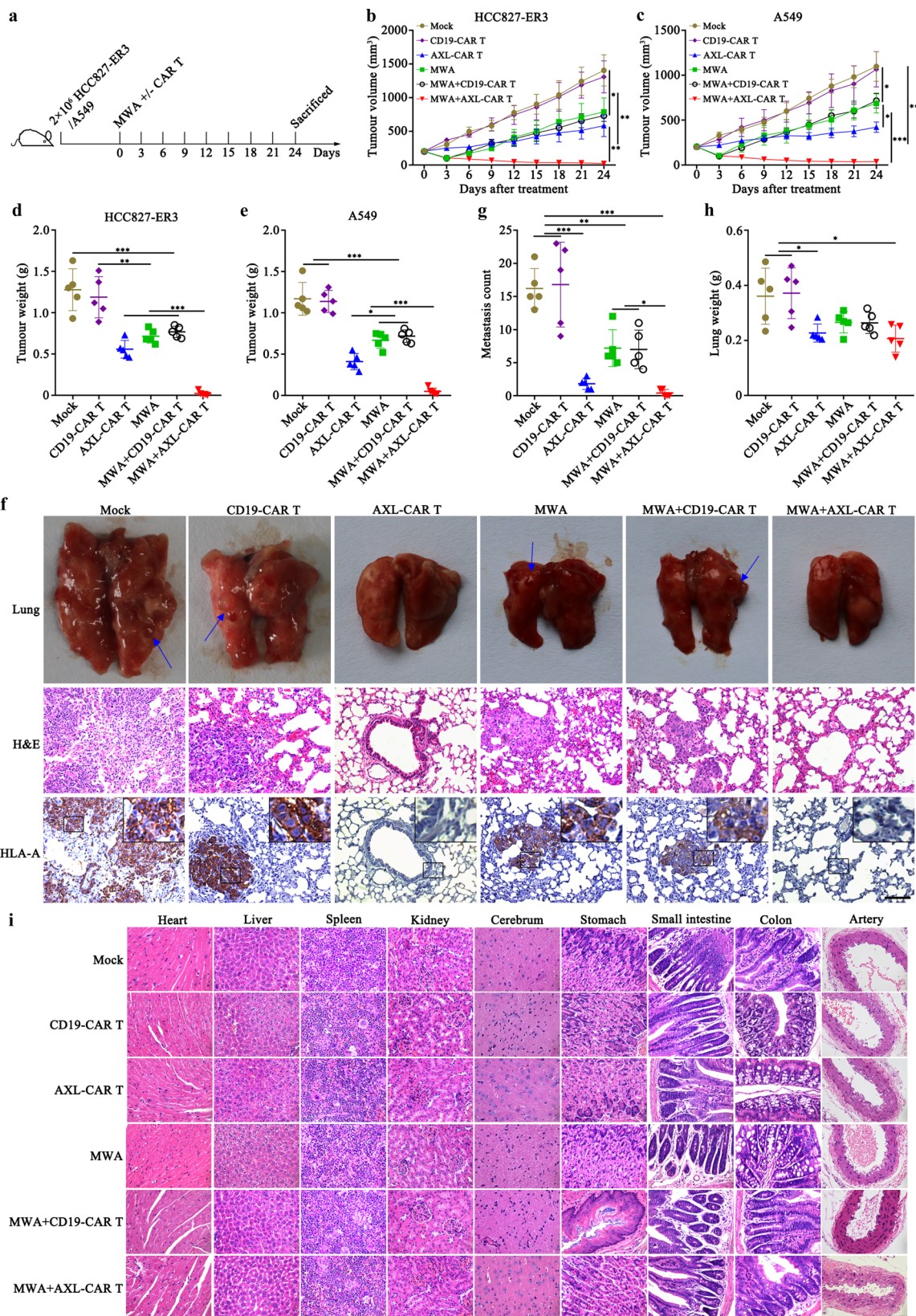

damage was observed in the organs from mice treated with MWA and/or AXL-CAR T cells by HE staining (Fig. 3i), and there were no significant differences in blood biochemical indexes between the treatment and control groups (Supplementary Table 4), which indicated a good safety threshold when considering AXL-CAR T cells can target AXL protein in both human and mouse tissue. Finally, in a local and distant tumour model, we further demonstrated that combination therapy improved tumour suppression in distant site through promoting CAR T cell infiltration (Supplementary Fig. 8). Together, compared with monotherapy, MWA combined with AXL-CAR T cells exhibited superior local and systemic CDX (cell-derived xenografts) tumour regression under a good safety threshold in NSG mice.

**Fig. 3 | MWA promotes antitumour activity of AXL-CAR T cells against lung cancer. a** Schematic representation of the experiment examining effects of a combination of MWA and CAR T cell administration ($n = 5$ mice per group). **b/c** Tumour volume of mice subcutaneously injected with HCC827-ER3 or A549 cells. **d, e** Mean tumour weight in each group at the end point. **f** Detection of HLA-A expression in harvested lung tissues from each group. Representative images are shown ($n = 5$ per group). Positive staining for HLA-A was observed in lung tissues from mock, CD19 and/or MWA groups. Blue arrow indicated metastatic nodules.

**g** Counted metastasis in lungs of each group. **h** The weight of lungs from each group. **i** Histopathological analysis of murine organ tissues. Different tissues were harvested, formalin-fixed, paraffin-embedded, and stained using haematoxylin and eosin (HE) staining. Representative staining image fields (magnification ×400) are shown ($n = 5$ per group). Scale bars represent 100 μm (**f**) or 50 μm (**i**). Data are presented as mean ± SD (**b–e, g, h**) and analysed by two-way ANOVA (**b, c**) or one-way ANOVA (**d, e, g, h**) with Tukey's multiple comparisons test. *$p < 0.05$, **$p < 0.01$, ***$p < 0.001$. Source data and exact $p$ values are provided as a Source Data file.

## MWA promotes the infiltration of AXL-CAR T cells in an AXL-positive NSCLC PDX tumour model

To investigate whether MWA could enhance the tumour-suppressive properties of AXL-CAR T cells in NSCLC, we constructed an AXL-positive NSCLC PDX (patient-derived xenografts) tumour model for an in vivo combination study to mimic the primitive TME of NSCLC (Fig. 4a, b). When tumour volume reached about 200 mm³ on day 30, we randomly assigned these mice to four groups: mock, MWA, AXL-CAR T cells and MWA combined with AXL-CAR T cells. We observed that administration of AXL-CAR T cells alone moderately suppressed tumour growth, which could be due to the hypofunction of CAR T cells in the immunosuppressive microenvironment. Furthermore, MWA monotherapy failed to suppress tumour growth because of the tumour residue. In contrast, compared with the outcomes of monotherapies, MWA combined with AXL-CAR T cells efficiently suppressed tumour growth and led to better tumour regression at the endpoint (Fig. 4c). These beneficial outcomes were accompanied by higher infiltration of CAR T cells (comb-CAR T cells) in both the peripheral blood on days 42 and 54 and harvested tumours than that in the AXL-CAR T group (mono-CAR T cells) (Fig. 4d–f), despite the dosage of CAR T cells that was twice higher in the latter group. Importantly, the numbers of CD8⁺ CAR T cells, known as a major population of tumour-killing cells, were higher owing to MWA in the combination group than in the AXL-CAR T monotherapy group (Fig. 4g, h). These findings confirmed that comb-CAR T cells had significantly higher homing and antitumour properties in a NSCLC PDX tumour model.

## MWA potentiates the accumulation and function of AXL-CAR T cells in NSCLC PDX tumours by remodelling TME

To explore how MWA enhances the accumulation and killing activity of AXL-CAR T cells in NSCLC PDX tumours, we analysed several key limiting factors in the TME. We sacrificed five mice in each group on days 33 and 42, respectively, and harvested the tumours for further investigations (Fig. 4a). High levels of hyaluronic acid (HA), the main component of extracellular matrix in TME, constitute a physical barrier that limits the access of immune cells and impedes the intratumoural vasculature[29]. Therefore, we examined the HA content in PDX tumours on day 33. Notably, HA expression was more abundantly detected in the mock and AXL-CAR T groups, whereas in the MWA and combination groups, the HA content was lower on day 33 (Fig. 5a). This was confirmed by ELISA results, as more HA protein was detected in the mock or AXL-CAR T group than that in the MWA or combination group (Fig. 5b). The underlying mechanism of HA depletion may be attributed to heat, because we observed downregulation of HA levels in NSCLC cell lines after incubation at higher temperatures for 5 min (Supplementary Fig. 9). Moreover, higher blood flow rate (Fig. 5c/d) were detected in the MWA or combination groups than that in the mock or AXL-CAR T groups. The monitored blood flow rate reached peak in 12 h after MWA by laser speckle equipment (Fig. 5d). These observations were further verified by immunofluorescence staining (Fig. 5e), as tumours in the combination group displayed the greatest amount of vasculature on day 42, which might be advantageous for CAR T cell infiltration and supply of nutrients. Furthermore, tumour interstitial fluid pressure (IFP) was reduced in the combination and MWA groups after the treatment compared to that in the mock or AXL-CAR T groups (Fig. 5f). Notably, decreased IFP is also regarded as a

favourable feature for tumour permeability and T cell infiltration[30]. Most importantly, elevated partial pressure of $O_2$ ($PO_2$) in the combination group was detected after MWA on day 42 (Fig. 5g). This observation is of great importance, because hypoxia strongly influences antitumour immune responses, which results in the hypofunction of CAR T cells[5]. These results suggested that MWA remodelled the TME in NSCLC PDX tumours by reducing HA, increasing vascularization, decreasing IFP, and alleviating hypoxia. Collectively, these phenomena enhanced the accumulation and function of AXL-CAR T cells in NSCLC PDX tumour model.

## MWA induces the activation and memory phenotype of tumour-infiltrating AXL-CAR T cells and reduces their exhaustion in PDX tumours

To further explain the enhanced antitumour ability of AXL-CAR T cells in vivo, we sacrificed 5 mice in the combination and AXL-CAR T groups on day 33 and 42 and sorted tumour-infiltrating CAR T cells to perform a series of tests. We found that the comb-CAR T cells expressed higher levels of the activation markers CD69 and CD95 than mono-CAR T cells (Fig. 6a), which indicated that MWA activated the infiltrating CAR T cells. Subsequently, we analysed the cytokine profile of bulk tumours. We found that the levels of cytokines, including IL-2, IFN-γ and TNF-α, were very low in tumours treated with AXL-CAR T cell monotherapy (Fig. 6b), suggesting that mono-CAR T cells were hypofunctional and/or that the absolute number of AXL-CAR T cells responding to the tumour cells was low. On the other hand, higher levels of IL-2, IFN-γ and TNF-α were detected in comb-CAR T cells than those in mono-CAR T cells (Fig. 6b). We also confirmed the same trend in serum, having found that low but detectable systemic levels of cytokines were produced after the combination treatment (Supplementary Fig. 10). These results suggested that MWA activated the infiltrating CAR T cells.

Recent studies have shown that the development of memory characteristics is associated with increased antitumour efficacy and persistence in adoptively transferred T-cell subsets[31]. Surprisingly, the expression of the Tcm markers CD62L⁺CD45RO⁺ was significantly upregulated in the comb-CAR T cells on day 42, compared to that in mono-CAR T cells (Fig. 6c–e). We also confirmed that the percentage of CD25⁺FoxP3⁺ Tregs in CD4⁺ TILs was low and that there was no significant difference between the mono- and comb-CAR T cells (Fig. 6f). In contrast, the comb-CAR T cells exhibited lower levels of exhaustion markers, such as PD1, TIM3 and CTLA4, than the mono-CAR T cells (Fig. 6g). Together, these data indicated that the comb-CAR T cells displayed higher cytokines release, more activation/memory differentiation and less exhaustion than mono-CAR T cells. Although it is not clear from these data that the functionality of the tumour-infiltrating CAR T cells was directly changed by the MWA, we believe that enhanced tumour control may have been mediated by either improvement of function or increased CAR T cell infiltration into tumours induced by MWA effects.

## The modified TME resulting from MWA reprograms AXL-CAR T cell metabolism

Because elevated oxygen levels were observed after MWA in PDX tumours, to study the metabolic effects of increased oxygenation in tumour-infiltrating AXL-CAR T cells, we measured oxygen consumption rate (OCR) and extracellular acidification rate (ECAR) in sorted

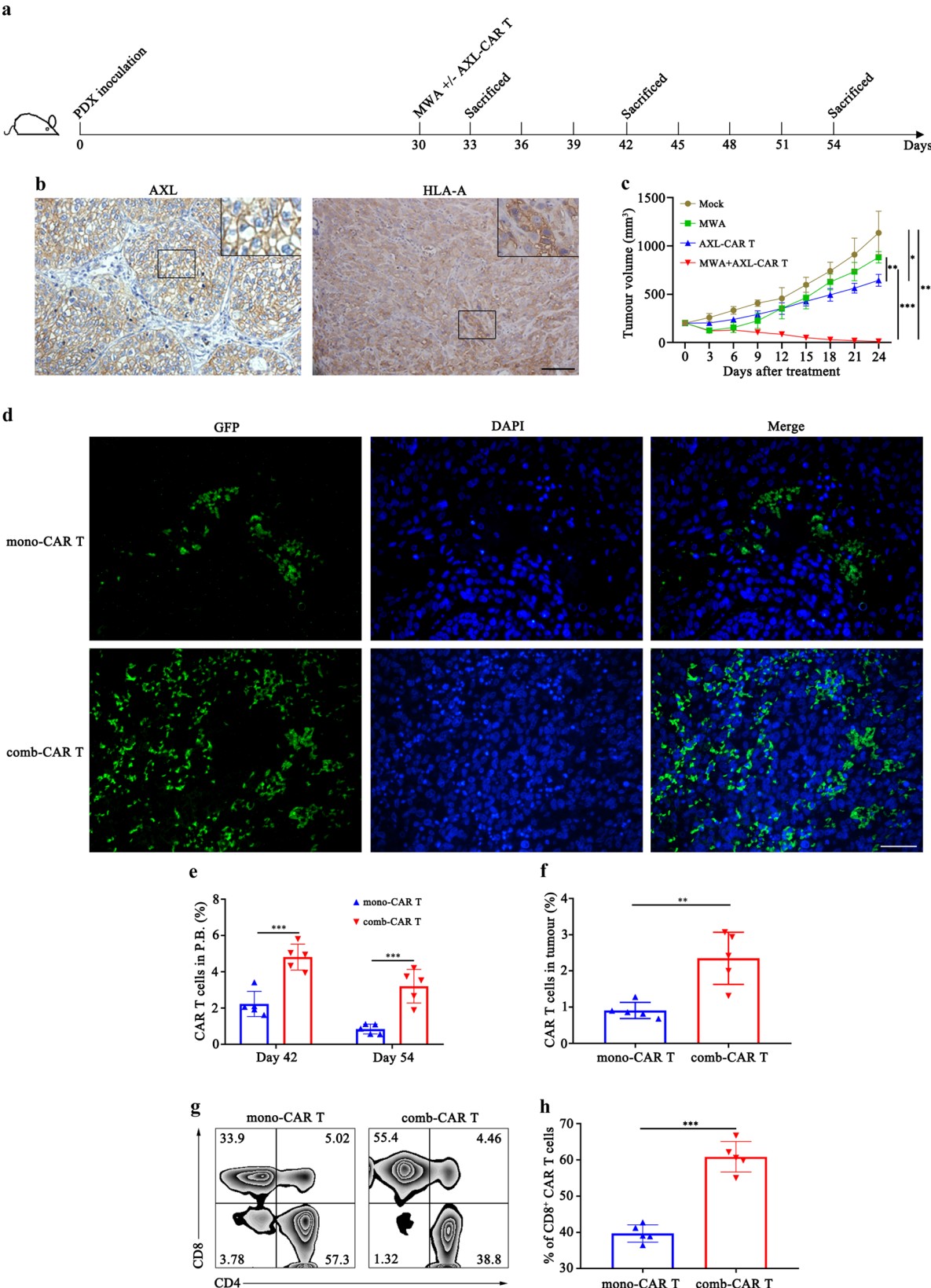

AXL-CAR T cells on day 42 by using a Seahorse Bioanalyzer. Intriguingly, there was a robust increase in the relative ATP levels and basal and maximal respiratory capacities of comb-CAR T cells compared with those in mono-CAR T cells following decoupling of the mitochondrial membrane using FCCP (Fig. 7a–d). In contrast, the ECAR of infiltrated CAR T cells in tumours was elevated in mono-CAR T cells in

comparison to that in comb-CAR T cells (Fig. 7e–h). Moreover, there was a substantial increase in the number of mitochondria in comb-CAR T cell mass, compared with those in mono-CAR T cells (Fig. 7i), which indicated increased mitochondrial biogenesis and oxidative metabolism in comb-CAR T cells. These oxidative features suggested that comb-CAR T cells were metabolically distinct from mono-CAR T cells,

**Fig. 4 | Combined treatment with MWA and AXL-CAR T cells promotes infiltration of CAR T cells and suppresses NSCLC PDX tumour growth. a** Schematic diagram showing the schedule of treatment of NSCLC PDX bearing mice. **b** Detection of AXL and HLA-A in the patient-derived-xenografts (PDX) tumour by IHC. Representative staining image fields (magnification ×200) are shown ($n = 5$ per group). Scale bar represents 100 μm. **c** Tumour volume curve in different groups. Data are analysed by two-way ANOVA with Tukey's multiple comparisons test (Mock vs AXL-CAR T $p = 0.02$, Mock vs MWA + AXL-CAR T $p = 0.001$, MWA vs AXL-CAR T $p = 0.001$, MWA/AXL-CAR T vs MWA + AXL-CAR T $p < 0.0001$). **d** Representative immunofluorescence staining images of GFP-marked CAR-T cell infiltration into cancer tissues from the NSCLC PDX bearing mice. Formalin-fixed, paraffin-embedded mouse tissue sections were stained for GFP (green) and DAPI (blue). Representative staining image fields (magnification ×200) are shown ($n = 5$ per group). Scale bar represents 100 μm. **e, f** Percentage of CAR T cells in the peripheral blood on days 42 and 54 and in the tumour on day 42 of PDX bearing mice determined by flow cytometry (**e**: day 42: mono-CAR T vs comb-CAR T $p < 0.0001$. day 54: mono-CAR T vs comb-CAR T $p = 0.0001$. **f**: mono-CAR T vs comb-CAR T $p = 0.003$. two-sided unpaired $t$-test). **g, h** Representative proportions of CD8+ tumour-infiltrating CAR T cells measured by flow cytometry (**h**: mono-CAR T vs comb-CAR T $p < 0.0001$, two-sided unpaired $t$-test). Data are obtained from 5 biologically independent samples per group and presented as the mean ± SD (**c, e, f, h**). *$p < 0.05$, **$p < 0.01$, ***$p < 0.001$. Source data are provided as a Source Data file.

with the former displaying a greater capacity for oxidative metabolism and the latter exhibiting glycolytic metabolism.

To gain insight into the mechanism of metabolic changes conferred by the elevated oxygen supply, we measured the expression of candidate genes implicated in glycolysis. Strikingly, *GLUT1, PDK1, PGK1*, and *G6PD*, the main enzymes implicated in glycolytic metabolism, were expressed at significantly lower levels in comb-CAR T cells than those in mono-CAR T cells (Fig. 7j). In contrast, we observed significantly higher levels of *CPT1A* and *FABP5* mRNA in comb-CAR T cells than in mono-CAR T cells. This finding further illustrated that comb-CAR T cells obtained a greater capacity for oxidative metabolism in modified TME after MWA, which might be due to enhanced central memory differentiation, because memory T cells use oxidative phosphorylation to meet energy demands[32] (Fig. 6c–e).

Taken together, these findings suggested that comb-CAR T cells had increased mitochondrial oxidative metabolism and decreased expression of glycolysis-related genes compared to mono-CAR T cells.

### Combination treatment is effective and safe in humanized immunocompetent mice

Previous study indicated that AXL might expressed in innate immune cells, and AXL inhibition could reprogram AXL-expressing innate immune cells to affect antitumour immunity[33,34]. Therefore, to further assess the efficacy and safety of the combination strategy, we performed combination therapy in humanized immunocompetent mice (huNOG-EXL). A549 cells were subcutaneously inoculated into the flank of huNOG-EXL mice. When the tumour volume reached about 200 mm³, mice were randomly allocated to four groups: mock (PBS), MWA, AXL-CAR T cells (i.v.), MWA combined with AXL-CAR T cells (i.v.) (Fig. 8a). Neither AXL-CAR T cells nor MWA monotherapy was able to suppress tumour growth. Administration of AXL-CAR T cells combined with MWA exhibited effective antitumour response (Fig. 8b). Moreover, the tumour-infiltrated T cells were measured. We found that combination treatment resulted in higher donor (GFP+) and host T cell infiltration in the tumours than monotherapies (Fig. 8c/d). These results highlight the importance of combination treatment in an immunocompetent setting, which possibly activates endogenous adaptive and innate antitumour activity. Macrophages were critical components in innate antitumor immunity[33]. To assess whether combination therapy alters potential macrophage-mediated immune suppression, we analyzed the phenotypes of macrophages in tumours. We found that MWA led to mild elevation of total macrophages (CD68+) without significant difference in all groups (Fig. 8e). Combination or AXL-CAR T treatment induced downregulation of CD206+ expression from CD68+ macrophages in tumours (Fig. 8f), which is consistent with decreased M2 polarization. In contrast, MWA did not induce downregulation of CD206+ expression of macrophages (Fig. 8f).

We further evaluated the safety of AXL-CAR T cells in huNOG-EXL mice. There was no significant difference among all groups in terms of leukocyte percentages both in tumour and spleen, including T (CD3+), B (CD19+), NK (CD56+) and monocyte cells (CD14+) (Fig. 8g/h). Moreover, no obvious damage was observed in the organs from humanized

NOG-EXL mice treated with MWA and/or AXL-CAR T cells by HE staining (Fig. 8i). Together, these results indicated that combination treatment was effective with low toxicities in humanized immunocompetent mice.

## Discussion

The application of CAR T cells for the treatment of solid tumours faces a unique set of challenges. The major hindrance for CAR T cell immunotherapy in NSCLC is the selection of a specific antigen that can discriminate tumour from normal tissues[35]. Furthermore, homing of CAR T cells to the tumour sites must be sufficient and effective. After having infiltrated into the tumour, CAR T cells have to survive, proliferate, persist, and exert cytotoxic action in the hostile immunosuppressive TME[4]. Moreover, infiltrated CAR T cells should differentiate and persist as memory T cells that confer long-term protection[31]. To induce a favourable antitumour outcome, CAR T cells have to fulfil all these aforementioned complicated tasks.

High AXL expression in 76% (32/42) of pancreatic ductal adenocarcinoma was reported, especially present in invasive cells[23]. In the present study, we observed high AXL expression in lung cancer tissues, whereas its expression in twenty-two kinds of normal tissues was limited. We therefore hypothesised that it could be a potential target for CAR T cells directed against NSCLC. As expected, AXL-CAR T cells exhibited intrinsic target-dependent cytotoxic activity and cytokine secretion in vitro. Moreover, moderate antitumour efficacy of AXL-CAR T cells was observed in both subcutaneous and lung metastatic CDX tumour models in vivo. In addition, CAR T cells incorporating an scFv originating from YW327.6S2 as the antigen-binding element had no toxicity towards normal mouse tissues, although the antibody could bind strongly to the mouse AXL protein[24]. These observations confirmed that AXL could be a valuable and safe target for CAR T immunotherapy against NSCLC.

MWA is an important technique in the treatment of inoperable lung cancer. MWA can reprogram TME and boost local and systemic immune responses[11,12]. In the present study, to enhance the function of CAR T cells and treat NSCLC CDX and PDX bearing mice, we applied a treatment regimen by combining MWA with the administration of AXL-CAR T cells. The PDX tumour model maintained the heterogeneity of primary NSCLC tumours and proved to be an excellent tool for cancer research, as the tumours were directly transferred from patients to NSG mice[36]. This setting allowed us to faithfully mimic the microenvironment of the primary tumour. Notably, the combination of MWA and AXL-CAR T cells yielded a synergistic effect, which enhanced T cell killing capability and prevented distant metastasis in CDX tumour models. Furthermore, the combination treatment displayed superior antitumour efficacy and increased infiltration of CAR T cells in both the tumour and peripheral blood in the PDX tumour model. It has been clearly shown previously that the number of CAR-expressing tumour-infiltrating lymphocytes is directly correlated with their efficacy in solid tumours[8,9]. Thus, it is reasonable that augmentation of CAR-expressing tumour-infiltrating lymphocytes by MWA in this study enhanced antitumour efficacy. Several Phase I/II studies are underway to evaluate

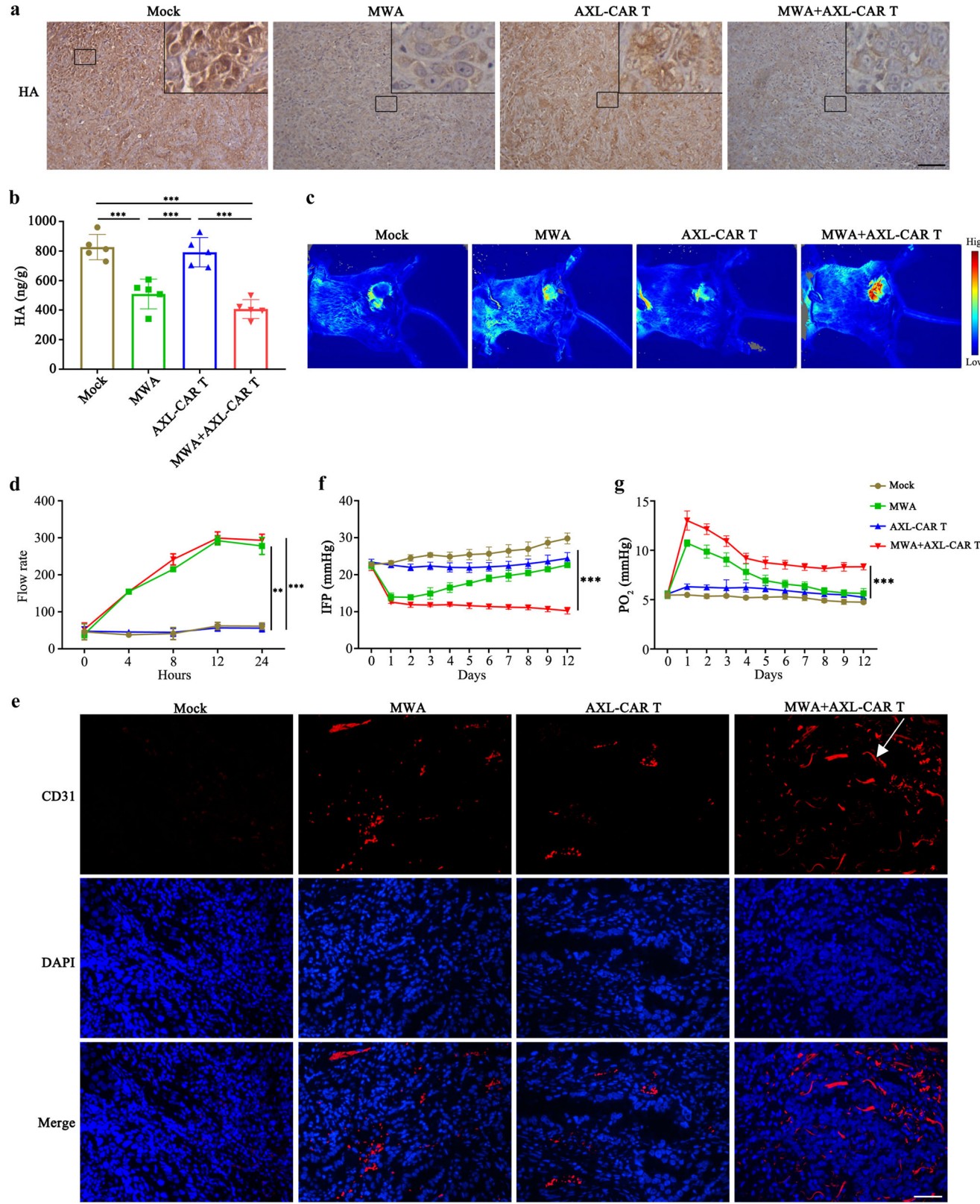

Type I/II AXL kinase inhibitors in combination with chemotherapy or immune checkpoint inhibitors in various solid tumours, however, the overall antitumour efficacy are far from satisfaction[28]. Our data were consistent with the reports that photothermal therapy or cytokine-armed oncolytic adenoviruses facilitated the accumulation and effector function of CAR T cells within solid tumours[8,9,37]. Moreover, our combination strategy may show advantages to photothermal combination therapy for deep tumours because of the good penetration and accessibility of MWA needles[10]. What's more, considering that MWA is a frequently-used method in lung cancer, suggesting that our combination strategy has high translational promise in clinical practice.

We further investigated the mechanisms underlying the effects of combination therapy by focusing on the TME-modifying effects of MWA. It has been reported that advanced solid tumours are largely

**Fig. 5 | TME remodelling following MWA in NSCLC PDX tumours. a, b** HA detection in different groups on day 33 by IHC (magnification ×200) and ELISA, respectively. Scale bar: 100 µm. Representative staining image fields with similar results are shown. Data are presented as the mean ± SD over 5 biologically independent samples per group and analysed by one-way ANOVA with Tukey's multiple comparisons test (**b**: Mock vs MWA $p = 0.0002$, Mock/AXL-CAR T vs MWA+AXL-CAR T $p < 0.0001$, MWA vs AXL-CAR T $p = 0.0006$). **c, d** Representative blood flow rate images (each group $n = 3$) and curve around the tumour region at 0, 4, 8, 12, and 24 h after MWA treatment by laser speckle equipment (d: MWA vs Mock $p = 0.004$, MWA vs AXL-CAR T $p = 0.003$, MWA+AXL-CAR T vs Mock $p = 0.0007$, MWA + AXL CAR T vs AXL-CAR T $p = 0.0005$). **e** Immunofluorescence staining of blood vessels (CD31) in tumour tissues collected from mice after treatment on day 42 (magnification ×200). Representative staining image fields with similar results are shown ($n = 5$ per group). Scale bar: 100 µm. **f, g** IFP and $PO_2$ measurements in mice before and after the treatment (Mock/MWA/AXL-CAR T vs MWA+AXL-CAR T $p < 0.0001$). Data are presented as the mean ± SD ($n = 3$ mice per group) and analysed by two-way ANOVA with Tukey's multiple comparisons test (**d, f, g**). *$p < 0.05$, **$p < 0.01$, ***$p < 0.001$. Source data are provided as a Source Data file.

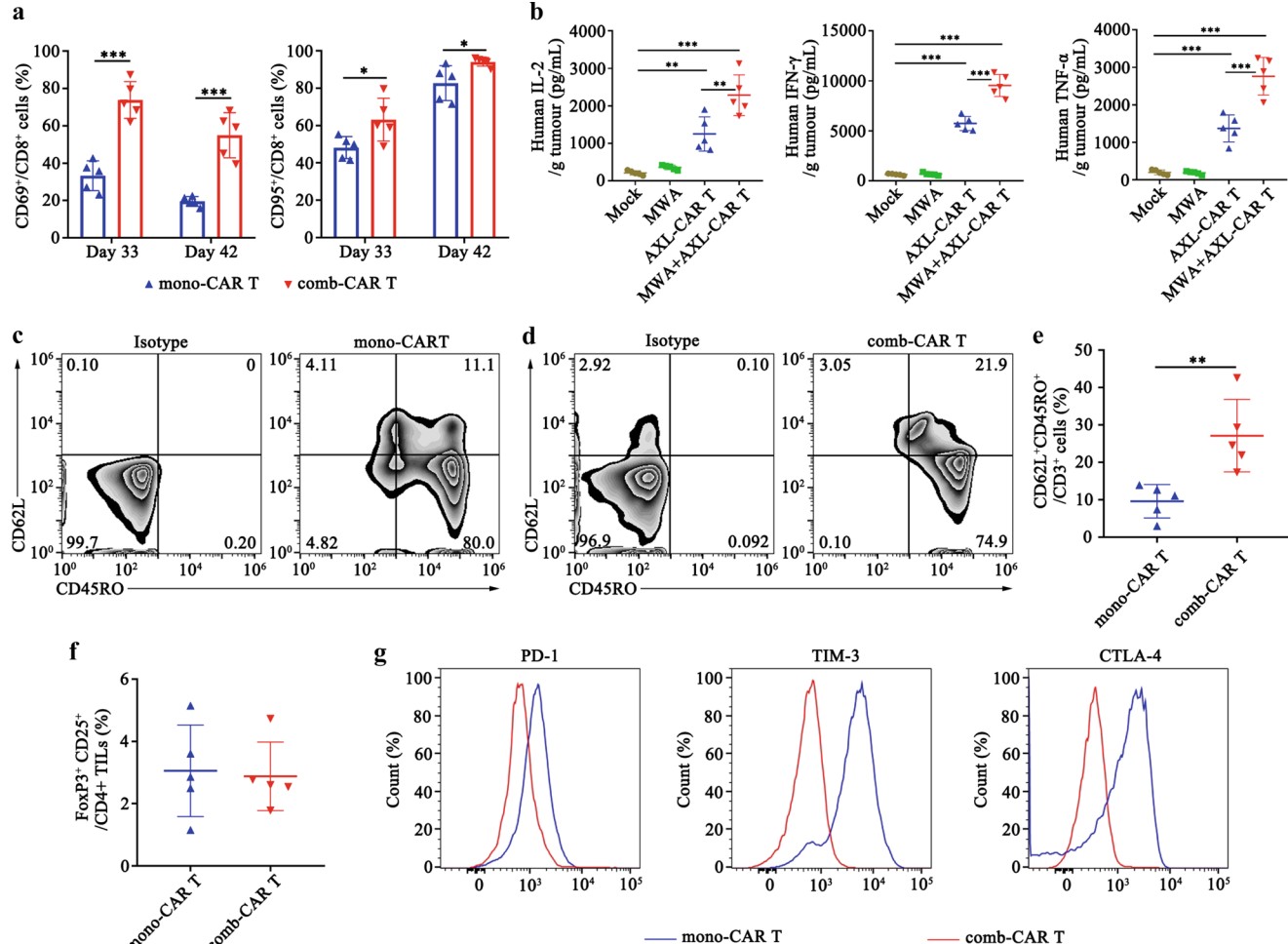

**Fig. 6 | MWA enhances activation and memory phenotype of tumour-infiltrating AXL-CAR T cells and reduces their exhaustion in PDX bearing mice. a** Expression of activation markers CD69 and CD95 in tumour-infiltrating CD8+ T cells on day 33 and 42 analysed by flow cytometry. Data were analysed by two-sided unpaired *t*-test (CD69: day 33: mono-CAR T vs comb-CAR T $p < 0.0001$. day 42: mono-CAR T vs comb-CAR T $p = 0.0002$. CD95: mono-CAR T vs comb-CAR T $p = 0.03$). **b** Cytokine profile of bulk tumours on day 42. Cytokines in the homogenate supernatant were analysed by ELISA. Data were analysed by one-way ANOVA with Tukey's multiple comparisons test (IL-2: Mock/MWA vs MWA+AXL-CAR T $p < 0.0001$, AXL-CAR T vs MWA+AXL-CAR T $p = 0.002$. IFN-γ/ TNF-α: Mock/MWA/ AXL-CAR T vs MWA + AXL-CAR T $p < 0.0001$). **c–e** Expression of CD62L/CD45RO in tumour-infiltrating CAR T cells on day 42 after cell treatment. Data are analysed by two-sided unpaired *t*-test (**e**: mono-CAR T vs comb-CAR T $p = 0.006$). **f** Analysis of Treg infiltration into the tumours by flow cytometry. Data are analysed by two-sided unpaired *t*-test. **g** Representative expression of exhaustion markers in tumour-infiltrating CAR T cells on day 42 after cell treatment is shown, as determined by flow cytometry. Data are obtained over 5 biologically independent samples per group (**a–g**) and presented as the mean ± SD (**a, b, e, f**). *$p < 0.05$, **$p < 0.01$, ***$p < 0.001$. Source data are provided as a Source Data file.

composed of extracellular matrix and immunosuppressive modulators, and are characterised by aberrant vascularisation, resulting in hypoxia and altered nutrient availability[6]. This prompted us to attempt to overcome the hostile immunosuppressive environment by using MWA in combination with CAR T cells. Intriguingly, we found that MWA remodelled the TME in NSCLC PDX tumours by reducing HA, increasing vascularisation, decreasing IFP, and alleviating hypoxia. These favourable changes that enriched AXL-CAR T cells and enhanced

their functional properties have been previously reported in preclinical tumour models[29,38]. HA and IFP have been shown to impart increased barrier integrity and resistance to drug delivery and immunotherapy[30]. Moreover, HA and IFP reduction in this study resulted in better tumour vascularisation, permeability and T cell infiltration, which further ensured enrichment of CAR T cells in the tumour. Furthermore, we suppose that MWA induced the following effects. First, the proinflammatory chemokines and cytokines released

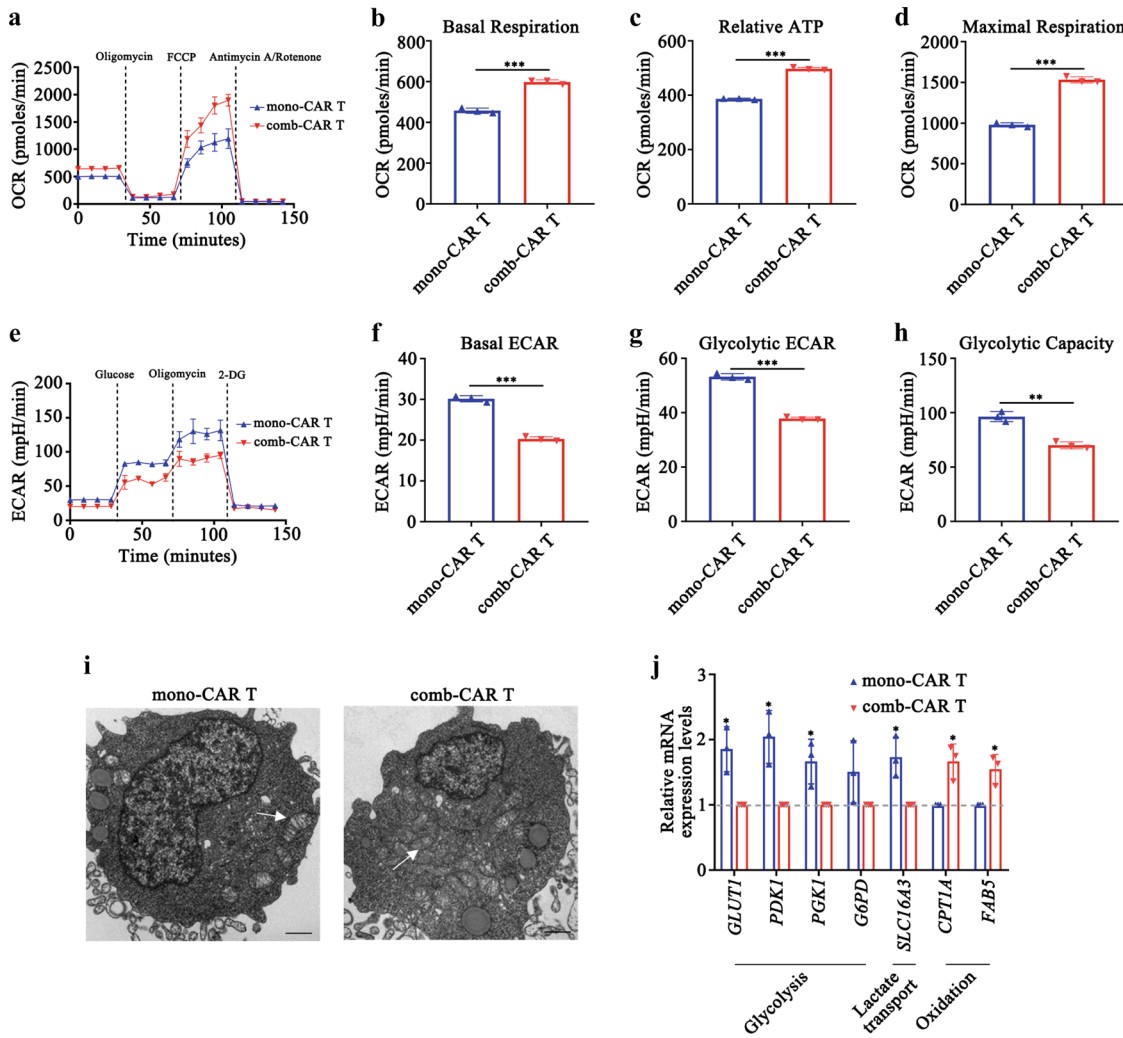

**Fig. 7 | MWA reprograms AXL-CAR T cell metabolism. a–d** Representative oxygen consumption rate (OCR) curves and mitochondrial stress test quantifications at different metabolic states (**b**: mono-CAR T vs comb-CAR T $p = 0.0001$. **c, d**: mono-CAR T vs comb-CAR T $p < 0.0001$). **e–h** Representative extracellular acidification rate (ECAR) curves and glycolysis stress test quantifications at different metabolic states (**f, g**: mono-CAR T vs comb-CAR T $p < 0.0001$, **h**: mono-CAR T vs comb-CAR T $p = 0.001$). **i** Transmission electron microscopy of mono- and comb-CAR T cells imaged on day 42, white arrow indicates mitochondrion. Representative image fields with similar results are shown ($n = 3$ per group). Scale bars represent 1 μm. **j** Relative mRNA expression levels of genes involved in glycolytic metabolism and lipid oxidation assessed in mono- and comb-CAR T cells (GLUT1, PDK1, SLC16A3, CPT1A, FAB5: mono-CAR T vs comb-CAR T $p = 0.01$, PGK1: mono-CAR T vs comb-CAR T $p = 0.03$). Data are obtained over 3 biologically independent samples per group and presented as the mean ± SD (**a–h, j**). Data are analysed by two-sided unpaired $t$-test (**b, d, f–h, j**). *$p < 0.05$, **$p < 0.01$, ***$p < 0.001$. Source data are provided as a Source Data file.

from the ablated tissue or tumour cells, as well as the disruption of local extracellular matrix and tissue components, provided a local inflammatory response, which might have generated specific immunity, promoted CAR T cell activity and eventually enhanced CAR T cell infiltration. Second, tumour antigens released from tumour cells destroyed by MWA could also have enhanced the killing potency of CAR T cells because early and direct antigen exposure benefits the activation and lytic capability of CAR T cells[39]. All these factors may have directly or indirectly resulted in the increased infiltration, accumulation, and activation of CAR T cells. These ideas are inline with the fact that comb-CAR T cells expressed higher levels of the activation markers CD69/CD95 and cytokines than mono-CAR T cells. Our findings support the idea that MWA created a relatively permissive environment in PDX bearing mice, which enhance the activation, accumulation, persistence, and antitumour efficacy of tumour-infiltrating CAR T cells in vivo.

TME hypoxia in solid tumours is attributed to the unrestrained growth of tumour cells and anomalous angiogenesis. Tumour hypoxia has attracted increasing attention as it poses a considerable obstacle to

drug delivery and decreases the therapeutic efficacy of chemotherapy, radiotherapy, and immunotherapy[40]. Tumour-infiltrating CAR T cells have substantial challenges to survive and perform effector functions in a hypoxic TME. A previous in vitro study revealed that hypoxia impaired the expansion, differentiation, and cytokine production of CAR T cells[41]. These effects may decrease the ability of CAR-T cells to eradicate solid tumours. In the present study, we noticed that $PO_2$ in tumours that underwent combination treatment was elevated after MWA. This change, potentially accompanied by increased nutrient supply via increased vasculature, may contributed to the augmented capacity to generate mitochondrial mass, and eventually provided a survival advantage[42]. Previous studies have suggested that pre-processed hypoxic CTLs or CAR T cells exhibit better antitumor ability[43–46]. Our results seem contrary to these findings because MWA induced relatively higher $PO_2$ levels. However, it is reasonable and consistent when considering that either mono- or comb-CAR T cells were maintained in hypoxic conditions[47,48]. That is, we did not reverse hypoxia but alleviated hypoxia in the NSCLC PDX tumours with MWA. This change may be beneficial for CAR T cells to perform their function. Unfortunately,

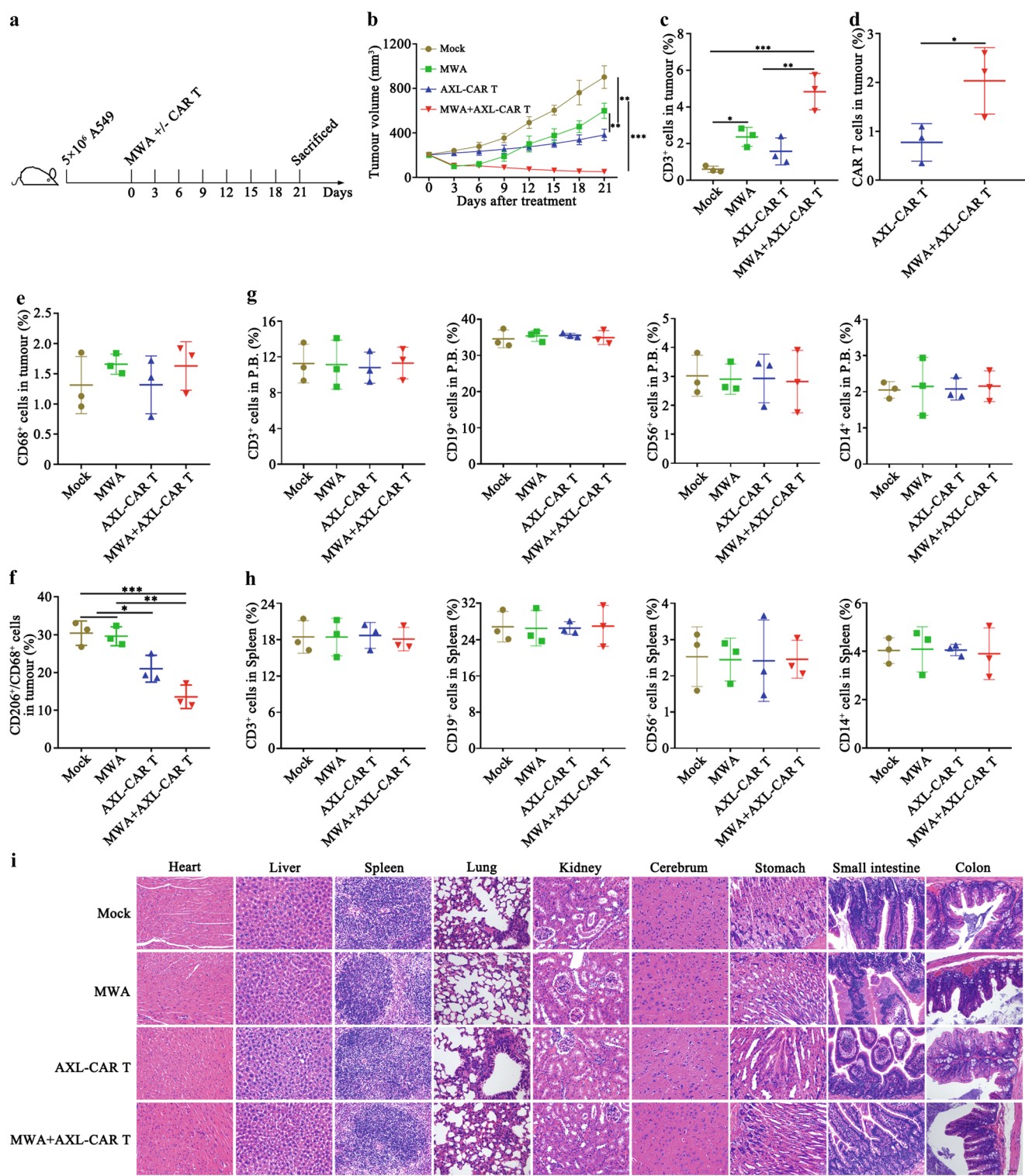

the mechanism underlying the enhanced comb-CAR T cell function induced by $PO_2$ fluctuation (and which $O_2\%$ may be better) is still unknown. We suppose that $PO_2$ fluctuation is not an independent factor that changes comb-CAR T cell properties, it might be the complex TME modifications induced by MWA that potentiate the comb-CAR T performance and ultimately result in better tumour regression. Clearly, this is an important area requiring future detailed, comprehensive analysis.

We consistently observed that comb-CAR T cells had a higher mitochondrial oxidative respiratory capacity, which has been shown to be an important characteristic of memory T cell development, but

whether changes in metabolism drive memory differentiation, or whether memory differentiation drives oxidative metabolism is still unknown[32]. More importantly, mono-CAR T cells had higher expression of PD1, TIM3, and CTLA4 than comb-CAR T cells, which could be a possible reason why mono-CAR T cells were not effective in PDX-bearing mice. Thus, metabolic reprogramming of comb-CAR T cells by MWA may not only confer preferential reliance on oxidative phosphorylation as the predominant energy source for meeting the metabolic demands required for enhanced CAR T cell function, but also support central memory differentiation and T cell persistence.

**Fig. 8 | Antitumour efficacy and safety of MWA+/- AXL-CAR T cells against A549-derived tumour in humanized immunocompetent mice. a** Schematic representation of a combination of MWA and AXL-CAR T cell administration. Briefly, humanized NOG-EXL immunocompetent mice received a subcutaneous injection of $5 \times 10^6$ A549 cells in the flank. When tumour volume reached about 200 mm³, mock, MWA+/- AXL-CAR T cells (i.v.) were administrated against A549-bearing mice. **b** Tumour volume was measured every three days ($n = 5$ per group). Data are presented as the mean ± SD and analysed by two-way ANOVA with Tukey's multiple comparisons test (Mock vs MWA $p = 0.004$, MWA vs AXL-CAR T $p = 0.003$, Mock VS AXL-CAR T $p = 0.0002$, Mock vs MWA+AXL-CAR T $p = 0.001$, MWA vs MWA+AXL-CAR T $p < 0.0001$, AXL-CAR T vs MWA+AXL-CAR T $p = 0.0002$). The percentages of total CD3⁺ T cells (**c**) and CAR T cells (GFP⁺) (**d**). The percentages of total macrophages (**e**) and M2 macrophages (**f**) by flow cytometry (**c**: Mock vs MWA $p = 0.049$, Mock vs MWA+AXL-CAR T $p = 0.0003$, MWA vs MWA+AXL-CAR T $p = 0.009$, AXL-CAR T vs MWA+AXL-CAR T $p = 0.002$. **d**: AXL-CAR T vs MWA+AXL-CAR T $p = 0.049$. **f**: Mock vs AXL-CAR T $p = 0.03$, Mock vs MWA+AXL-CAR T $p = 0.0008$, MWA vs AXL-CAR T $p = 0.04$, MWA vs MWA+AXL-CAR T $p = 0.001$). The leukocyte percentage in peripheral blood (**g**) and spleen (**h**) was detected by flow cytometry. Data are analysed by one-way ANOVA with Tukey's multiple comparisons test (**c**, **e**–**h**) and two-sided unpaired *t*-test (**d**). **i** Histopathological analysis of murine organ tissues by haematoxylin and eosin (HE) staining ($n = 5$ per group, magnification ×200). Scale bar represents 50 μm. Data are obtained over 3 biologically independent samples and presented as the mean ± SD (**c**–**h**). *$p < 0.05$, **$p < 0.01$, ***$p < 0.001$. Source data are provided as a Source Data file.

AXL is expressed by innate immune cells, including macrophages, and participates in immune responses. Systemic AXL inhibition may enhance the efficacy of cancer therapy via multiple mechanisms[49]. Thus, we further evaluated the efficacy and safety of combination strategy in humanized immunocompetent mice. Combination therapy effectively suppressed tumour growth than monotherapies. We observed combination treatment enhanced infiltration of non-CAR host T cells and adoptively transferred donor CAR T cells. Moreover, AXL-CAR T cells may modulate the tumour immunosuppression by reducing M2 polarization of tumour-associated macrophages (TAM). Given the evidence that M2-like macrophages exert pro-tumour effects by suppressing T cell infiltration and cytotoxic T cell function[50], this reprogramming of the TME is expected to enhance the efficacy of AXL-specific CAR T cell therapy. The similar observations were reported that AXL receptor inhibition decreased M2 characteristics of TAM[33]. However, the detailed mechanism of how AXL-CAR T cells influence the macrophages or other AXL-expressing innate immune cells and in what degree is still uncovered in the present study. Considering that MWA effects are highly associated with infiltration of transferred donor AXL-CAR T and host T cells, and in turn, the infiltrated AXL-specific CAR T cells may have the capability in suppress the innate immunosuppressive immune cells in TME, it may be interesting to determine the synergistic therapeutic relevance of the combination strategy in the future study.

Meanwhile, potential "on-target/off-tumour" effects of AXL-CAR T cells will be considered when they are further applied in clinic. Individualized AXL expression profiles should be examined to ensure safety when considering the utilisation of AXL-CAR T cells in the clinical setting. Moreover, the immunodeficient NSG mice used in this study lack a completely functional immune system. Lack of an intact immune system in NSG mice may overlook the important immunological aspects of combination therapy. Nevertheless, NSCLC PDX tumour model used here may mimic the TME of primary NSCLC to some extent. We further assessed the efficacy and safety of the combination therapy in humanized immunocompetent NOG-EXL mice. Combination treatment seems to be effective without observed toxicities in humanized immunocompetent mice.

In summary, we characterised AXL as a target antigen for CAR T cells in human NSCLC. Then we described a combination therapy with MWA and AXL-CAR T cells for the treatment of NSCLC. AXL-CAR T cells alone only moderately suppressed tumour growth, but their combination with MWA enabled effective AXL-CAR T cell therapy that reduced HA, increased vascularisation, decreased IFP, and alleviated hypoxia. These effects, in turn, may favourably enhance the activation, accumulation, survival, persistence, and killing capacity of AXL-CAR T cells in vivo. Our results provide a rationale for combination therapy with MWA and CAR T cells, which might have translational value for the treatment of NSCLC in the clinical setting. Phase I clinical trial to further evaluate the safety and efficacy of AXL-CAR T cells in advanced AXL-positive lung cancer patients (NCT03198052, ClinicalTrials.gov) in our institution is currently underway.

## Methods

### Ethics approval
All human sample collections and analysis were approved by the Institutional Review Board of the Second Affiliated Hospital of the Guangzhou Medical University. The informed consent was obtained by human participants. Animal experiments were performed with the approval of the Second Affiliated Hospital of Guangzhou Medical University Experimental Animal Care Commission.

### Determination of AXL expression by IHC
IHC procedures were performed following standard protocols as previously described[51]. AXL expression levels in 90 paraffin-embedded NSCLC tissue sections (including 40 EGFR TKI-resistant cases, Supplementary Table 2) and 22 kinds of non-tumour paraffin-embedded sections (Supplementary Fig. 1, Supplementary Table 1) were detected by staining with a rabbit anti-AXL antibody (1:500, Cell Signalling Technology, #8661, clone C89E7) and assessed by an experienced pathologist. A 4-point scale was used to define different AXL expression levels in patient samples. Score 0 indicated no AXL expression; scores+, ++, and +++ indicated weak, medium and strong AXL expression levels, respectively. The percentages of AXL-positive samples with different scores were also recorded.

### Cells lines and cell culture
A549, HCC827, and HEK-293T cell lines were obtained from the American Type Culture Collection. The erlotinib-resistant HCC827-ER3 cell line was established at the Case Western Reserve University (Cleveland, Ohio, USA) as previously described[14]. Cells were maintained in a humidified atmosphere of 95% air and 5% $CO_2$ at 37 °C in DMEM (Gibco) or RPMI-1640 (Gibco) supplemented with 10% foetal bovine serum (Gibco), 100 IU/mL penicillin (Gibco), and 100 IU/mL streptomycin (Gibco). The Free-Style 293-F cells were obtained from Invitrogen and cultured in Freestyle 293 expression medium (GIBCO). For in vitro cytotoxicity and in vivo imaging experiments, A549, HCC827 and HCC827-ER3 cells were lentivirally transduced with the pWPXLD-Luc(+)/eGFP virus harbouring genes encoding GFP and luciferase (GL) (Supplementary Fig. 3e). All cells were routinely tested for mycoplasma contamination.

### Animals
Six- to eight-week-old female NSG (NOD-*Prkdc*ˢᶜⁱᵈ*Il2rg*ᵗᵐ1/Bcgen, Biocytogen, Beijing, China) mice and the fourteen to sixteen-week-old female humanized NOG-EXL mice (NOD.Cg-Prkdc scid Il2rg tm1SugTg(SV40/HTLV-IL3,CSF2) 10-7Jic/JicTac) expressing human GM-CSF and human IL-3, engrafted with CD34 + human cord blood stem cells (Taconic Biosciences)[52,53] were housed under specific pathogen-free conditions at a constant temperature ($22 \pm 0.5$ °C) and humidity ($60 \pm 2$%) under an automatically controlled 12/12 h light/dark cycle, and were provided autoclaved food and water at the Experimental Animal Centre of the Second Affiliated Hospital of Guangzhou Medical University (Guangzhou, China). Tumour volume was measured every three days with a caliper and calculated with the

following equation: Tumour volume = (length×width$^2$)/2. The maximal tumour size over 2000 mm$^3$ is forbidden by the ethics committee.

## IFP, tumour PO$_2$, and tumour blood flow measurements after MWA

To detect TME alterations, several key factors influencing the TME were monitored. IFP and PO$_2$ were assessed by the wick-in-needle technique and polarographic electrode (POG-203; Unique Medical) before (day 0) and after MWA (days 1–9, 12), respectively, as previously described[30]. Then, tumour blood flow rate was monitored around the tumour region at 0, 4, 8, 12 and 24 h after MWA procedure using the laser speckle equipment (MoorFLPI-2, Moor Instruments).

## In vivo studies in CDX and PDX models

For the cell line-derived NSCLC subcutaneous xenograft models, $2 \times 10^6$ A549 or HCC827-ER3 cells in 100 μL of PBS were subcutaneously injected into the right flank of NSG mice on day 0. When tumour nodes reached about 50 mm$^3$, the mice were divided into three groups of six mice each: mock, CD19-CAR T, and AXL-CAR T, and received PBS or $1 \times 10^7$ CAR T (CD19- or AXL-CAR T) cells intravenously. On day 39 (A549) and 45 (HCC827-ER3) after tumour inoculation, all mice were sacrificed. Tumour volume was measured every 3 days.

To construct pulmonary metastasis models, $1 \times 10^6$ A549 GL cells in 100 μL of PBS were injected into NSG mice i.v., on day 0. Two weeks after the injection of tumour cells, the mice were subjected to BLI. Then mice were randomly divided into three groups of five mice each: mock, CD19-CAR T, and AXL-CAR T. Mice were administered PBS (mock) or $1 \times 10^7$ effector cells (CD19- or AXL-CAR T) suspended in 100 μL of PBS i.v. on day 14. Mice were monitored by using BLI.

To assess antitumour efficacy and safety of MWA (Vision-China Medical Devices R&D Centre) combined with the administration of AXL-CAR T cells, subcutaneous tumour models were established by inoculating HCC827-ER3, A549, HCC827 cells or AXL-positive NSCLC tissue in NSG mice or injecting A549 cells in humanized immunocompetent NOG-EXL mice. The experiment was initiated when the mean tumour volume reached about 200 mm$^3$. The mice were then randomly allocated to six groups for CDX models: mock (PBS), CD19-CAR T cells ($10^7$ cells, i.v.), AXL-CAR T cells ($10^7$ cells, i.v.), MWA (10 W, 40 s), MWA (10 W, 40 s) combined with CD19-CAR T cells ($10^7$ cells, i.v.), MWA (10 W, 40 s) combined with AXL-CAR T cells ($10^7$ cells, i.v.); or four groups for PDX and immunocompetent models: mock, MWA, AXL-CAR T cells, MWA combined with AXL-CAR T cells. CAR T cells were intravenously injected after MWA. Tumour volume was measured every 3 days. TME modification was evaluated before and after treatment. Blood was collected for biochemistry and CAR T cell infiltration analysis. Cytokine secretion in the peripheral blood and tumour tissue was measured by using ELISA or the Luminex assay (R&D Systems). Antibodies against human CD31 (1:1000, Cell Signaling Technology, #3528, clone 89C2), GFP (1:800, ab290, Abcam), C1QBP (1:1600, Cell Signaling Technology, #6502, clone D7H12), HLA-A (1:100, Abcam, ab52922, clone EP1395Y) and Ki-67 (1:500, Abcam, ab231172, clone SP6) were used for IHC, ELISA or immunofluorescence analyses in CDX or PDX tumour slices. All important mouse organs, including heart, liver, spleen, kidney, cerebrum, stomach, small intestine, colon and artery, were harvested, fixed with 4% paraformaldehyde and stained with haematoxylin-eosin (HE). For humanized immunocompetent NOG-EXL mice, the T cells and macrophages in tumours and the main leukocyte percentage in peripheral blood and spleen were detected.

## Bioluminescence imaging

Animals were anaesthetised with isoflurane and imaged using a cooled CCD camera system (IVIS 100 Series Imaging System, Xenogen) following an intraperitoneal injection of D-luciferin (Cayman Chemical) at a dose of 75 mg/kg. Average emissions were quantified using Living Image software (LI4.5.5, Xenogen).

## Isolation of tumour-infiltrating CAR T cells

Mice were sacrificed, and the tumours were collected, homogenised, and dissociated using the MACS Miltenyi Mouse Tumour Dissociation Kit (Miltenyi Biotec) according to the manufacturer's instructions. Purified CAR T cells were obtained by sorting (magnetic beads, Miltenyi Biotec) based on the manufacturer's instructions. The phenotype of CAR T cell was detected via flow cytometry (ACEA Novocyte D2060R). Sorted CAR T cells were utilised for metabolism analysis and transmission electron microscopy.

## Flow cytometry

All samples were analysed using a NovoCyte™ flow cytometer (ACEA Novocyte D2060R), and FlowJo software (FlowJo V10, TreeStar, Ashland, OR). The antibodies used included anti-AXL-PE (Thermo Fisher Scientific, #12-1087-42, clone DS7HAXL), anti-human CD3-FITC (Biolegend, #300406, clone UCHT1), anti-human CD4-APC (Biolegend, #300514, clone RPA-T4), anti-human CD8a-PE (Biolegend, #300908 clone HIT8a), anti-human TIM-3-PE (BD, #563422, clone 7D3), anti-human CTLA-4 PercP-Cy5.5 (Biolegend, #369607 clone BNI3), anti-human PD1-APC (Biolegend, #329908, clone EH12.2H7), anti-human CD95-APC (Biolegend, 305612, clone DX2), anti-human CD25-PE (Biolegend, #985802 clone M-A251), anti-human CD3-APC-CY7 (Biolegend, #317342 clone OKT3), anti-human CD4-PerCP (Biolegend, #300528, clone RPA-T4), anti-human CD8-PE-CY5 (Biolegend, #344769, clone SK1), anti-human CD25-APC (Biolegend, #302609, clone BC96), anti-human CD69-APC-CY7 (Biolegend, #310913, clone FN50), anti-human FOXP3-PE (Biolegend, #320107 clone 206D), anti-human CD45RO-PerCP/Cy5.5 (Biolegend, #304222, clone UCHL1), anti-human CD62L-PE (Biolegend, #304806, clone DREG-56), anti-human CD68-APC (Biolegend, #333810, clone Y1/82 A), anti-human CD19-PE (Biolegend, #302208 clone HIB19), anti-human CD14-PE-CY7 (Biolegend, #301814, clone M5E2), anti-human CD56-APC-CY7 (Biolegend, #362512 clone 5.1H11), anti-human CD206-PE (Biolegend, #321106, clone 15-2), mouse IgG1 kappa isotype control-PE (Biolegend, #400112, clone MOPC-21), mouse IgG1 kappa isotype control-FITC (Biolegend, #400108, clone MOPC-21), mouse IgG1 kappa isotype control-APC (Biolegend, #400120, clone MOPC-21), mouse IgG2a kappa isotype control-PerCP/Cy5.5 (Biolegend, #400252, clone MOPC-173), mouse IgG2a kappa isotype control-APC-CY7 (Biolegend, #400230, clone MPC-173), mouse IgG1 kappa isotype control-PerCP (Biolegend, #400148, clone MOPC-21), mouse IgG1 kappa isotype control-PE-CY5 (Biolegend, #400118, clone MOPC-21), mouse IgG2b kappa isotype control-APC (Biolegend, #400322, clone MPC-11), mouse IgG2a kappa isotype control-PE-CY7 (Biolegend, #400232, clone MOPC-173), mouse IgG1 kappa isotype control-APC-CY7 (Biolegend, #400128, clone MOPC-21). All antibodies were diluted 1:50 for use. FACS-related staining procedures were performed on ice for 30 min, and the cells were then washed with PBS containing 1% foetal bovine serum before cytometry analysis. Peripheral blood and tumour samples from mouse xenografts were treated with red blood cell lysis buffer (Thermo Fisher Scientific), and the cells were stained with the corresponding antibodies.

## Metabolism assays

ECAR and OCR were measured using a XFe96 Extracellular Flux Analyser (Seahorse Bioscience) with the Seahorse Cell Energy Phenotype Test Kit, as previously described[45]. Duplicates for sorted tumour-infiltrating CAR T cell sample from each mouse ($n = 3$) were performed. ECAR and OCR were measured simultaneously, before and after an injection of oligomycin (1 mM) and FCCP (1 mM).

For transmission electron microscopy, the sorted tumour-infiltrating CAR T cells were fixed with 2% glutaraldehyde and 1% osmium tetroxide. Cell samples were then cut into ultrathin sections, stained with 2% uranyl acetate, dehydrated, embedded, and stained with lead citrate. Images were acquired with a transmission electron microscope (FEI Tecnai G2 Spirit Biotwin, Eindhoven, the Netherlands).

## Western blotting

For western blotting, tumour and normal tissue samples or cell lines were washed with PBS and directly lysed by an ultrasonic cell disrupter (Sonics) in the Laemmli buffer (Bio-Rad Laboratories, USA). Lysates were electrophoretically separated on an 8% or 10% gradient SDS-PAGE gel (Bio-Rad Laboratories) and transferred to a nitrocellulose membrane. Subsequent procedures were modified from the standard protocol. Membranes were probed with corresponding primary antibodies, including anti-human AXL (1:1000, Cell Signalling Technology, #8661, clone C89E7), anti-human CD3 (1:25, Abcam, ab16669, clone SP7), C1QBP (1:1000, Cell Signaling Technology, #6502, clone D7H12).

## Statistics

Data are presented as the mean ± standard deviation or ± standard error of the mean. Differences between groups were analysed by *t*-test, one- or two-way ANOVA analysis of variance with Bonferroni *post hoc* tests. Grey-scale analysis of WB images was performed by ImageJ (V1.8.0). The Kruskal-Wallis test was utilised to compare the non-normally distributed endpoints. For survival data, Kaplan-Meier curves were plotted and compared using the log-rank test. GraphPad Prism 8.0 was used for statistical calculations. Effects were considered statistically significant if $p < 0.05$.

## Reporting summary

Further information on research design is available in the Nature Research Reporting Summary linked to this article.

## Data availability

All the data are available within the Article, Supplementary Information or Source Data file. Source data are provided as a Source Data file. Source data are provided with this paper.

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

## Acknowledgements

This research was supported by the National Natural Science Foundation of China (no. 81672276 and No. 81872069 to Zhenfeng Z., no. 81872613 and no. 81573142 to L.Y., no. 82102860 to S.Y.). Guangzhou Basic and Applied Basic Research Foundation (no. 202201010880 to B.C., no. SL2023A04J00530 to M.L.). The National Key R&D Program of China (2019YFA0904400 to Q.Z.), Shenzhen Science and Technology Project (SGDX2020110309280301 to Q.Z.), the Science and Technology Development Fund, Macau SAR (File no. FDCT/0004/2019/AFJ, FDCT/0043/2021/A1 and FDCT/0002/2021/AKP to Q.Z.), University of Macau (File no. MYRG2019-00069-FHS and MYRG2022-00143-FHS to Q.Z.), and the PhD Start-up Fund of the Second Affiliated Hospital of Guangzhou Medical University to B.C. and M.L.. We would like to thank P.L. (University of Chinese Academy of science) for kindly endow the backbone lentiviral vector pWPXLd-2A-eGFP. We thank Vision-China Medical Devices R&D center for providing Microwave Ablation devices.

## Author contributions

B.C., M.L., H.L., L.Y., Q.Z., and Zhenfeng Z. conceived and designed the project and contributed to the interpretation of data. L.W., K.Z., M.C., X.C., Y.F., S.Y., S.F., C.Z., X. Ye., J.Z., Z.Z., X.Y., M.Z., Q.W. and L.X. contributed to the acquisition and analysis of data. B.C. drafted the manuscript. Zhenfeng. Z. and Q.Z. jointly supervised and revised the manuscript.

## Competing interests

The authors declare no competing interests.
