## [Peer Review File · Nature Communications]

Remodelling of tumour microenvironment by microwave ablation potentiates immunotherapy of AXL-specific CAR T cells against non-small cell lung cancerREVIEWER COMMENTS

Reviewer #1 (Remarks to the Author): with expertise in CAR-T, metabolism, immunosuppression

The paper is very interesting and written in simple and clear way with a good rational flow. It cannot be denied by the experiments that the combination of anti-AXL CAR-T with MWA has a very effective ability in vitro and in vivo to target and kill lung tumour. Despite the positive judgment this paper needs some changes to be ready for a publication.

Results Comments:

1. The figure 1a shows the absence of expression of AXL in non tumour tissue, which is a control of figure 1b, so for this reason should be moved in the supplementary section.
2. Figure 1b is the most relevant panel, more tissue samples should be showed in this figure, 4 representative samples are very limited. If the samples are part of TMA (tumour Tissue microarray) all the array should be added in the supplementary.
3. AXL staining on primary lung tumour should be provided as the authors show that AXL staining is very bright on tumour cell line in figure 1F.
4. Supplementary figure 2S: the staining for CD3 in the 3 donors is unnecessary and not informative. The key figure is the expression of GFP after transduction before and after enrichment.
5. In Figure 3a-c different number of AXL-CART were injected, the group that received only CART has been injected with 10^7 CART, instead the group that received the combined treatment received 5×10^6 AXL-CART. Although I understand that if higher number of CART don't effect the tumour grow, less cells will have the same of lower effect. But in term of experimental setting the two group should receive the same number of CART.
6. Figure 3e : the metastasis are not evident, the picture should be improved as quality and the metastasis count should be reported for each mice analysed.
7. Figure 3h: all the mice staining need to be added if not in the main figure the authors should add in the supplementary. It is difficult to believe that the low expression of Ki-67 is associated with low proliferative cells and not due to the absence of tumour, because there is not a double staining with tumour cell marker.
8. Figure 6b should be described in text after figure 6a, the description of the panels figure need to be consecutive.
9. All the panel in figure 7 are not in the consecutive order, the authors describe panel 7a and 7b after the description of the panel 7k and 7J. The authors also in this case they need to change or the order of panel or the flow description in the text.
10. The OCR and ECAR graphics missing the description of experimental samples, if the T cell from each mouse has been run individually and how many replicas for each T cell sample. All these information should be added to the legends or to the method section of METABOLISM ASSAYS.

Additional comments:

- ALL the figure legends missing how many time the experiment in each panel have been repeated.
- In the introduction the authors don't describe the AXL molecule in the context of literature (small paragraph should be added)
- The English should be re-edited.

Reviewer #2 (Remarks to the Author): with expertise in AXL and cancer

General remarks

The investigators confirmed that AXL can be highly expressed in a subset of lung tumors. Based on this information, they propose that this group of patients may benefit from anti-AXL CAR-T cell therapy. They have used various NSCLC preclinical models to assess the potential benefit of CAR-T cell therapy, and generated interesting data supporting some activity of Third generation AXL-CAR-T cells in the context of NSCLC. In addition, they provide evidence that combining MicroWave Ablation

(MWA) with AXL-CAR T approach could be an interesting to improve antitumor effects of CAR-T cells. This benefit appeared to correlate with modulation of the tumor microenvironment caused by MWA, better CART persistence and tumor infiltration.

The work is interesting and original in some aspects. It is felt however that it does not represent a significant advance in the field and there are numerous problems. although this study does confirm that AXL targeting using CART cells has therapeutic potential (previous work exists on various preclinical models), the place of such approach +/- combination remains unclear compared to ready-to use and less toxic approaches, or the use of other CART cells that are more advanced in their clinical development for solid tumors including in the setting of NSCLC (Niels Schaft <https://www.mdpi.com/2072-6694/12/9/2567>). Although the technological aspect is interesting, with data on different preclinical models, I do think it still requires significant improvement in terms of organization, and presentation of the data. The study on environmental changes triggered by MWA is interesting but perhaps too preliminary. Another important concern is that authors do not cite adequate references to related and previous work.

Therefore, there are many caveats and limitations in this study that makes it difficult to accept for publication.

Below are listed some specific concerns and suggestions for improvement

1-The potential of AXL-CAR-T have been published elsewhere with preclinical data on breast cancer models in vivo and various models in vitro. The authors completely eluded this literature. This point should be clarified. Authors should discuss previous work and discuss better how their work can bring to the field.

I found at least 2 references:

A novel AXL chimeric antigen receptor endows T cells with anti-tumor effects against triple negative breast cancers Jing Weia et al Cellular Immunology 331 (2018) 49–58.

<https://doi.org/10.1016/j.cellimm.2018.05.004>

Engineered IL-7 Receptor Enhances the Therapeutic Effect of AXL-CAR-T Cells on Triple-Negative Breast Cancer Zhenhui Zhao et al. doi: 10.1155/2020/4795171 .

<https://pubmed.ncbi.nlm.nih.gov/31998790/>

not that, AXL CAR-T have entered clinical trials among other CAR-T directed to “solid tumors”

<https://clinicaltrials.gov/ct2/show/NCT03198052>

2- Authors don't discuss and consider important roles played by AXL in non-cancer cells including neurons, endothelial cells, immune cells, including NK and myeloid cells, with roles in homeostasis, angiogenesis, neurogenesis, and innate immunity.

Haley Axelrod et al 2014 <https://www.ncbi.nlm.nih.gov/pmc/articles/PMC4253401/>

Martha Wium et al Cancers 2021, 13, 1521. <https://doi.org/10.3390/cancers13071521>

3- potential issue regarding the absence of AXL in normal tissues (tumor specificity): poor quality images, poor resolution, renders very difficult proper interpretation of the data Figure 1. Therefore, tumor specificity of AXL remains questionable. Figure 1: shows negative staining in a panel of normal tissues. I don't see evidence that AXL is not expressed in “non-tumour” tissues. Current Microphotographs (or table S1) are not convincing and do not provide sufficient resolution to appreciate if AXL is expressed by tumor cells and not by normal cells (and which cell types), tissues. Clarification is required on this aspect. As mentioned above, It is important in the context of the literature establishing that AXL can be expressed in multiple cell types like endothelial cells, and myeloid cells including macrophages as well as lymphocytes such as NK cells.

To be more convincing, instead of immunoblots, it would be nice to see immunostains with cancerous and adjacent non-tumor lung tissues (non-malignant epithelium, areas with vascular cells, immune infiltrates. increase magnification and quality for normal tissues, or provide additional images as supplementaries.

4- The choice of the CAR generation is unclear

It appears that authors have used 3rd generation CAR-T (CD28+41BB+CD3 endodomains) that are not entered in clinical practice. Why? 2nd generation CAR-T (2 endodomains) are approved in certain indications and known to be relatively safe in patients. Less is known about 3rd generation CAR-T cells.

For sake of clarity, authors should explain this choice, and whenever relevant, additionally include a comparison with 2nd generation CAR-T cells. For instance, a significant improvement of the significance would be to show that the combinatory MWA/CART approach has significant effects when integrating 3rd generation CAR-T, but more limited effects with 2nd AXL-CART cells.

5- A technical concern, the number of CAR-T infused in mice

The number of CART cells injected in mice is quite high (1×10^7). It is common to see 1 to 10×10^6 CAR-T injection in preclinical studies, and in comparison to patients typically receiving $1-10 \times 10^6$ per kg of bodyweight. Authors should clarify this point and if they have dose response data, they should include it.

6-Some additional analysis would be important to support the gain of adding MWA to CAR-T therapy:

-in light of their claim that MWA therapy could potentiate CAR-T effects. It would be keen for the authors to address the question whether MWA could help reduce the dose of CAR-T cells infused, thus limiting potential side effects and toxicity.

- Figure 3; as A549 seems to be a good model in Fig2 experiments. I was surprised that authors did not include A549 growth curves using MWA approach combined with CAR-T in Figure 3

-Figure 3; some control conditions with "CD-19 CAR-T +/- MWA" are missing to check tumor antigen specificity of the MWA effects. I wonder if MWA could also potentiate off-targets / side/toxicity effects.

-another interesting experiment would be to use AXL negative HCC827 xenograft to test the combination strategy.

7-for sake of clarity and in the interpretation of analysis, improvement is required Figure 4 and Figure 5: bad quality IHC stainings, luminosity. It would be great if one could discriminate lymphocytes from tumor cells. what type of HLA is stained (HLA ? A, B, C ?) Authors should correct figure labels.

8-(Figure 7A) Because the HIF-alpha has a very short-half time (approx. 5 mins), and because WB experiments require a certain number of cells, numerous investigators have reported difficulties in detecting HIF-alpha from purified lymphocytes. I wonder how authors generated such data. It's not clear if CAR-T cells were purified from the tumor bulk (from how many mice) before protein extraction? HIF-alpha expression could come from contaminating tumor fraction. More evidence is necessary to associate HIF deregulation in CAR-T and MWA.

I would suggest to attenuate statement, remove this data, or provide additional evidence that HIF/or HIF pathway is deregulated upon MWA treatment in CAR-T.

9- clarification required on the impact of MWA on TME and CART.

The reported mechanisms of the effect of MWA are still poorly defined and the cellular components involved unclear. For instance, it's not clear which CART subpopulations express the different markers (CD4+, CD8+, or regulatory-like CAR-T subsets?) Figure 6 and 7. It seems important yet to clarify this point, especially because the proposed relation with hypoxia is hard to interpret, with data at odds with recent reports.

Indeed, the authors made the assumption that hypoxia has deleterious effects on T cells, which contrast with recent reports that consistently showed improvement of CAR-T and T cells effector functions upon hypoxic stress. In fact, there is a possibility that the MWA effects are independent of hypoxia or involve complex metabolic loops. Thus, some 'toning down' of the conclusions on this aspect of the manuscript would be important, or additional evidence required in order to draw firm

conclusions."

10-Unfortunately, authors do not discuss the literature on this aspect and they should add relevant references like...

<https://doi.org/10.1172/JCI85834>

VHL-deficient CD8+ CAR T cells accumulate in tumors with an enhanced Trm-like phenotype. J Clin Invest. 2021;131(7):e143729 <https://doi.org/10.1172/JCI143729>

<https://doi.org/10.1016/j.ccell.2017.10.003>

<http://dx.doi.org/10.1016/j.celrep.2017.08.071>

11- related to my previous remark, Figure 6, CD25 is typically used as an activating mark of Tregs and could be a mark of Treg-like conversion of CAR-T. A major concern in the field. MWA seems to promote OxPhos metabolism and Treg are known to preferentially use OxPhos metabolism. it would be judicious to additionally include Foxp3 analysis and discriminate CD8+ and CD4+ populations. CD69 is a well-known marker of T cell activation, resident memory phenotype, also reported as an hypoxia-responsive gene. I would also suggest to analyze this marker

12- Figure 5h (Po2 measures): except at day 1, I don't see much difference. Is this significant ? As a side note, for non-expert, it's difficult to appreciate what pO2 level one should consider hypoxic or normoxic. Discussion and/or text could be expanded to include a bit more on this point.

13-The authors should improve discussion of the preclinical data and put them in context of the current knowledge and treatment strategies:

-potential benefit of their approach compared to typical combination therapy and what benefit compared to other AXL-targeting approaches. There are only 31 References and only 4 papers on AXL including their paper (14-17). No recent review or work discussing other AXL targeting strategies already used in clinics, or in clinical development. there are numerous recent reviews covering this aspect that authors could use. (see Tanaka, M. et al <https://doi.org/10.3390/ijms22189953> for update).

One question that arises is what would be the benefit of AXL-CAR-T compared to those and to what price in terms of toxicity.

-It's not clear for me why their method integrating AXL targeting would do better than any other CAR-T cells directed to solid tumors. (Niels Schaft <https://www.mdpi.com/2072-6694/12/9/2567>)

Reviewer #3 (Remarks to the Author): with expertise in CAR-T, lung cancer

This is a nice study which demonstrates that microwave ablation (MWA) can synergize with CAR-T cells to improve tumor control in xenograft models of NSCLC. The authors identify AXL as a potential CAR target that is expressed in a subset of NSCLC and shows limited expression in normal tissues, and develop AXL-specific CAR-T cells that show moderate activity in xenograft models of AXL+ NSCLC. MWA is shown to synergistically enhance tumor control by AXL CAR-T cells in a patient-derived xenograft model. Mechanistically, MWA is shown to increase blood flow, vascularization, and pO2 levels in tumors. Enhanced tumor control in the combination therapy group is associated with increased frequency of CAR-T cells in tumors, which show reduced HIF-1a expression, increased oxidative metabolism, and a more "central memory" phenotype.

Overall, the effect of MWA on CAR-T cell activity and tumor control is impressive and suggests that this is a promising combination therapy to pursue in the clinic. However, the authors make several claims about the mechanism of synergy between MWA and CAR-T cells that are not appropriately supported by the data. These claims either require more data to support, or should be modified:

1) The authors claim that “MWA induced the activation and memory phenotype, as well as reduced exhaustion of tumor-infiltrating AXL-CAR T cells in NSCLC PDX tumors.” It is not clear from the data that the functionality of tumor-infiltrating CAR T cells is changed by MWA. It is possible that enhanced tumor control is mediated primarily by increasing CAR T cell infiltration into tumors via enhanced vascularization/blood flow/etc rather than due to qualitative changes in CAR T cell function. MWA is shown to increase total cytokine levels in the tumor and blood, but this could simply be a product of the increased frequency of CAR T cells observed in the tumor and blood. Whether CAR T cells are more functional on a per-cell basis requires analysis of cytokine production by CAR T cells, either through intracellular cytokine staining or by sorting CAR T cells and examining cytokine production by ELISA/Luminex after restimulation ex vivo.

2) The authors claim that the “oxidative features in comb-CAR T cells most likely supported central memory differentiation and T cell persistence.” Although increased pO₂ levels in the tumor, increased oxidative metabolism in CAR T cells, and increased memory marker expression by CAR T cells are all observed, no data is provided to support a functional link between these phenotypes. Central memory T cells are known to exhibit greater oxidative metabolism, but whether changes in metabolism drive memory differentiation, or whether memory differentiation drives oxidative metabolism is not shown. Additionally, tumor burden/antigen load is known to strongly influence T cell differentiation and drive exhaustion, so it is possible that the enhanced memory phenotype of CAR T cells in MWA-treated tumors is simply due to the dramatically smaller tumors in these mice at this time point, and that metabolism of CAR T cells is altered simply as a result of memory differentiation.

3) The authors claim that “the combination therapy increased mitochondrial oxidative metabolism and decreased HIF-1a expression in tumor-infiltrating CAR T cells by alleviating TME hypoxia.” Although MWA is shown to increase tumor pO₂ levels and CAR-T cells show increased oxidative metabolism, no data is provided to support a functional link between these two observations. It is not shown whether enhanced oxidative metabolism is due to decreased hypoxia in the tumor, or whether it results indirectly from changes in tumor burden that preferentially induce memory differentiation of CAR T cells.

It is not necessary to provide data demonstrating a functional link between all these readouts, but these claims should be tempered to account for the fact that they are correlative.

Reviewer #4 (Remarks to the Author): with expertise in MWA

This manuscript reported that AXL-CART combined with MWA shows superior anti-tumor effect depending on a modification of TME. Overall, the work is of interest but there are many areas that need to be addressed. The major issues are listed below:

Introduction

1. a series of AXL-targeted small-molecule inhibitors have been investigated in clinic with favourable safety and pharmacokinetic profiles in NSCLC, why target AXL with CAR-T cells for the treatment of NSCLC should be explained, what's the potential advantage of AXL-CAR-T to R428 and BPI-9016M when combined with MWA.

2. The combinational value of MWA with CAR-T should be discussed, why achieved 1+1>2?

Methods

1. What's the sequence of MWA and CAR-T when combination therapy?

2. In the sixth part (page 21-22 MWA induces activation and memory phenotype of tumour-infiltrating AXL-CAR T cells and reduces their exhaustion in PDX tumours), the author showed that on days 42 ,when CAR-T cells had higher infiltration, they expressed more active markers. I think the same study should be carried out on the days 33 (3 days after treatment). To confirm whether AXL-CAR-T can change with time like HA or IFP.

3. Minor problem might be informal acronym, such as CDX and PDX, which were not explained before.

4. Study should add and compare the group AML-CAR-T and AML inhibitor to reflect the advantage of CAR-T that with potential side effects for solid tumor.

5. Does locoregional treatment will affect the expression of AXL in NSCLC? In other words, whether comb-CART treatment has the similar anti-tumor effects in AXL-negative NSCLC?

Results

1. In fig 4d, to show the infiltration of AXL-CART in the tumor tissue, however, it is not reasonable for indication of AXL-CART by using CD3 antibody (a T cell marker), a specific AXL antibody co-stain with CD8 was needed. MWA can induce inflammatory reaction, so the elevated CD3 may not be induced by AXL-CART.

2. In fig 4e, we notice that the number of CART was increased both in the PBMC and tumor tissue. However, the total cell mass of mono-CART group was twice more than it in the comb-CART group. Moreover, in their previous results, there are more AXL-CARTs were accumulated in the tumor region under comb-CART group. What was the reason for the expanding AXL-CART population?

Proliferation or cell death resistant?

3. Continue to question 3, although they provide sufficient evidence for supporting the hyperfunction of CART combined with MWA, given the cell amounts was more than twice in comb-CART (figure 4e), does the main reason that contribute to the upregulation of cytokines in combined group is associated with a functional enhanced AXL-CART or just due to the expanding AXL-CART population?

4. In Supplementary fig 4, the remaining tumor volume reduced sharply under the condition of 10W and 40s/10w and 80s, why don't choose a moderate parameter (10W and 20s)?

Discussion

1. In the last part of this work, they discuss that MWA can alleviate hypoxia in CART, which result in a decreased expression of T cell exhaustion markers. Does knockdown/knockout of Hif1 α can reverse the phenotype of mono-CART?

2. In discussion, they explore the limitation of AXL-CART for potential "on-target/off-tumour" effects, but it had been reported that the expression of AXL was ubiquitous (the protein level and mRNA level was shown in the below figures). In this paper, the AXL was mainly exist in the NSCLC, and hardly be detected in the normal tissues.

We thank all four reviewers for their helpful and professional comments for our manuscript. We strongly believe that we have addressed all the comments raised by the reviewers, as summarized below (Queries in black, response to Reviewers in blue). Totally, Figure 1/3/4/5/6/7 have been modified or updated, Figure S1/2/6/7 have been added. We request you to consider this article for publication in *Nature Communications*.

REVIEWER COMMENTS

Reviewer #1 (Remarks to the Author): with expertise in CAR-T, metabolism, immunosuppression

The paper is very interesting and written in simple and clear way with a good rational flow.

It cannot be denied by the experiments that the combination of ant- AXL CAR-T with MWA has a very effective ability in vitro and in vivo to target and kill lung tumour.

Despite the positive judgment this paper needs some changes to be ready for a publication.

Results Comments:

1. The figure 1a shows the absence of expression of AXL in non tumour tissue, which is a control of figure 1b, so for this reason should be moved in the supplementary section.

Response: Thank you so much for your comments and suggestion. We have moved **figure 1a** to the supplementary section as **figure S1 (line 1, page 48)**.

2. Figure 1b is the most relevant panel, more tissue samples should be showed in this figure, 4 representative samples are very limited. If the samples are part of TMA (tumour Tissue microarray) all the array should be added in the supplementary.

Response: We added more AXL positive primary lung tumour tissues in revised

figure 1a (line 9, page 7). These samples are not a part of TMA, but from regular pathological paraffin section.

3. AXL staining on primary lung tumour should be provided as the authors show that AXL staining is very bright on tumour cell line in figure 1F.

Response: Thanks for the advice. We provided those AXL positive samples from primary lung tumour tissues in revised **figure 1a (line 9, page 7).**

Fig. 1a Detection of AXL expression in NSCLC samples by IHC. Representative images (magnification $\times 400$) are shown. Scale bar represents 100 μm .

4. Supplementary figure 2S: the staining for CD3 in the 3 donors is unnecessary and not informative. The key figure is the expression of GFP after transduction before and after enrichment.

Response: Thanks! We agree that the key figure is the expression of GFP. We removed the unnecessary **figure S2** from the supplementary file.

5. In Figure 3a-c different number of AXL-CART were injected, the group that received only CART has been injected with 10^7 CART, instead the group that received the combined treatment received 5×10^6 AXL-CART.

Although I understand that if higher number of CART don't effect the tumour grow, less cells will have the same of lower effect. But in term of experimental setting the

two group should receive the same number of CART.

Response: We appreciate the advice. We uniformed the number of CAR T cells and re-performed the animal study in revised **figure 3**, both the CAR T or combination group received 10^7 CAR T cells (**line 1, page 13**).

6. Figure 3e : the metastasis are not evident, the picture should be improved as quality and the metastasis count should be reported for each mice analysed.

Response: We appreciate the comment. We improved the quality of previous figure 3e, which is now listed as current **figure 3f**. In the new **figure 3f**, nodules on the surface of the lung can be seen (**blue arrow**) and were verified by IHC staining. Moreover, we added metastasis counts from three types of slices (upper, middle and lower part of the lung). Counted data are shown in **figure 3g (line 1, page 13)**.

f, Detection of HLA-A expression in harvested lung tissues from each group. Positive staining for HLA-A was observed in lung tissues from the mock, CD19 and/or MWA groups. Blue arrow indicated metastatic nodules.

7. Figure 3h: all the mice staining need to be added if not in the main figure the authors should add in the supplementary. It is difficult to believe that the low expression of Ki-67 is associated with low proliferative cells and not due to the absence of tumour, because there is not a double staining with tumour cell marker.

Response: We agree and have added the photos of Ki-67 staining from three mice in

each group in **figure S7 (line 1, page 55)**. Indeed, there is a low expression tendency of Ki-67 in MWA combination group, probably due to significant growth suppression of the combined therapy.

Fig. S7 Ki-67 expression in tumours removed from mice of different groups in figure 3. Representative staining image fields (magnification $\times 400$) are shown. Scale bar, 100 μm .

8. Figure 6b should be described in text after figure 6a, the description of the panels figure need to be consecutive.

Response: Thanks for your advice. We have reorganized the order of **figure 6a** and **figure 6b** based on the description (**line 1, page 22**).

9. All the panel in figure 7 are not in the consecutive order, the authors describe panel 7a and 7b after the description of the panel 7k and 7J. The authors also in this case they need to change or the order of panel or the flow description in the text.

Response: Thanks for this critique. We have reordered and modified the panel of **figure 7** based on your advice (**line 10, page 24**).

10. The OCR and ECAR graphics missing the description of experimental samples, if the T cell from each mouse has been run individually and how many replicas for each T cell sample. All these information should be added to the legends or to the method section of MATABOLISM ASSAYS.

Response: We appreciate the suggestion. For MATABOLISM ASSAYS, duplicates for sorted tumour-infiltrating CAR T cell sample from each mouse (n=3) were performed, and this information has been added (**line 20-21, page 36**).

Additional comments:

- ALL the figure legends missing how many time the experiment in each panel have been repeated.

Response: Thank you so much for your advice. We have added the experimental repeat times in all the figure legends, The changes of figure legends are highlighted in yellow.

- In the introduction the authors don't describe the AXL molecule in the contest of literature (small paragraph should be added)

Response: We appreciate the comment. A paragraph was added to describe the AXL molecule in the contest of literature in the introduction section: "Because of AXL's discriminative expression and its implication in both tumour biology and chemotherapeutic resistance, a therapeutic that targets AXL could be a valuable cancer therapy. Although recent studies demonstrated that AXL may also play an important role in noncancer cells, including neurons, endothelial cells, and immune cells, with roles in homeostasis, angiogenesis, neurogenesis, and innate immunity that

may raise concern for potential haematologic and/or immune side effects, a series of AXL-targeted small-molecule inhibitors have been investigated in preclinical and clinical trials, including R428 and BPI-9016M, which showed favourable safety and pharmacokinetic profiles in NSCLC patients. Moreover, a high number of AXL-targeted monoclonal antibodies, such as hMAb173 for renal cell carcinoma, D9 and E8 for pancreatic cancer, and YW327.6S2 for NSCLC and breast cancer, have shown anticancer efficacy in preclinical and earlier clinical trial stages. In addition, AXL-targeted CAR T cells showed killing effects in breast cancer models *in vivo* and various tumour models *in vitro*. However, those monotherapies exhibited only moderate antitumour efficacy in previous studies, and none of the studies focused on the use of AXL-targeting CAR T cells for the treatment of NSCLC” (line 7-22, page 5; line 1-2, page 6, ref 17-28).

- The English should be re-edited.

Response: Thanks for your advice. We have re-edited this paper and **below is the language editing certificate.**

SPRINGER NATURE
Author Services Editing Certificate

This document certifies that the manuscript

Remodelling of tumour microenvironment by microwave ablation potentiates immunotherapy of AXL-specific CAR T cells against non-small cell lung cancer

prepared by the authors

Bihui Cao, Manting Liu, Yubo Zhou, Mingyue Cai, Lu Wang, Cheng Zhi, Xiaopei Chen, Yunfei Feng, Junping Li, Jinping He, Shuo Yang, Kangshun Zhu, Xin Yang, Yuling Hu, Hong Hu, Yu Tian, Ming Zhao, Qingde Wu, Linfeng Xu, Hui Lian, Lili Yang, Qi Zhao, Zhenfeng Zhang

was edited for proper English language, grammar, punctuation, spelling, and overall style by one or more of the highly qualified native English speaking editors at SNAS.

This certificate was issued on **April 28, 2022** and may be verified on the SNAS website using the verification code **94A8-97BC-7808-6F1A-4C36**.

Neither the research content nor the authors' intentions were altered in any way during the editing process. Documents receiving this certification should be English-ready for publication; however, the author has the ability to accept or reject our suggestions and changes. To verify the final SNAS edited version, please visit our verification page at secure.authorservices.springernature.com/certificate/verify.
If you have any questions or concerns about this edited document, please contact SNAS at support@as.springernature.com.

Reviewer #2 (Remarks to the Author): with expertise in AXL and cancer

General remarks

The investigators confirmed that AXL can be highly expressed in a subset of lung tumors. Based on this information, they propose that this group of patients may benefit from anti-AXL CAR-T cell therapy. They have used various NSCLC preclinical models to assess the potential benefit of CAR-T cell therapy, and generated interesting data supporting some activity of Third generation AXL-CAR-T cells in the context of NSCLC. In addition, they provide evidence that combining MicroWave Ablation (MWA) with AXL-CAR T approach could be an interesting to improve antitumor effects of CAR-T cells. This benefit appeared to correlate with modulation of the tumor microenvironment caused by MWA, better CART persistence and tumor infiltration.

The work is interesting and original in some aspects. It is felt however that it does not represent a significant advance in the field and there are numerous problems. although this study does confirm that AXL targeting using CART cells has therapeutic potential (previous work exists on various preclinical models), the place of such approach +/- combination remains unclear compared to ready-to use and less toxic approaches, or the use of other CART cells that are more advanced in their clinical development for solid tumors including in the setting of NSCLC (Niels Schaft <https://www.mdpi.com/2072-6694/12/9/2567>). Although the technological aspect is interesting, with data on different preclinical models, I do think it still requires significant improvement in terms of organization, and presentation of the data. The study on environmental changes triggered by MWA is interesting but perhaps too preliminary. Another important concern is that authors do not cite adequate references to related and previous work.

Therefore, there are many caveats and limitations in this study that makes it difficult to accept for publication.

Below are listed some specific concerns and suggestions for improvement

1-The potential of AXL-CAR-T have been published elsewhere with preclinical data

on breast cancer models in vivo and various models in vitro. The authors completely eluded this literature. This point should be clarified. Authors should discuss previous work and discuss better how their work can bring to the field.

I found at least 2 references:

A novel AXL chimeric antigen receptor endows T cells with anti-tumor effects against triple negative breast cancers Jing Wei et al. Cellular Immunology 331 (2018) 49–58.

<https://doi.org/10.1016/j.cellimm.2018.05.004>

Engineered IL-7 Receptor Enhances the Therapeutic Effect of AXL-CAR-T Cells on Triple-Negative Breast Cancer Zhenhui Zhao et al. doi: 10.1155/2020/4795171 .

<https://pubmed.ncbi.nlm.nih.gov/31998790/>

not that, AXL CAR-T have entered clinical trials among other CAR-T directed to “solid tumors” <https://clinicaltrials.gov/ct2/show/NCT03198052>

Response: Thanks a lot for your careful and helpful comments. We attempt to make improvement for this manuscript as following your suggestions.

We have added a paragraph of the application of AXL-CAR T cells in breast cancers in the introduction section: “In addition, AXL-targeted CAR T cells showed killing effects in breast cancer models in vivo and various tumour models in vitro. However, those monotherapies exhibited only moderate antitumour efficacy in previous studies, and none of the studies focused on the use of AXL-targeting CAR T cells for the treatment of NSCLC.” (line 19-22, page 5; line 1-2, page 6). We also cite two references as you mentioned (ref 26-27).

As for “AXL CAR-T have entered clinical trials among other CAR-T directed to “solid tumours” <https://clinicaltrials.gov/ct2/show/NCT03198052>”, we have to clarify that this clinical trail was just carried out in our institution (the Second Affiliated Hospital of Guangzhou Medical University), sponsored by Dr. Zhenfeng Zhang (corresponding author of this paper). This part was involved in last paragraph of the discussion section: “Phase I clinical trial to further evaluate the safety and efficacy of AXL-CAR T cells in advanced AXL-positive lung cancer patients (NCT03198052, ClinicalTrials.gov) in our institution is currently underway” (line 7-9, page 31).

2- Authors don't discuss and consider important roles played by AXL in non-cancer cells including neurons, endothelial cells, immune cells, including NK and myeloid cells, with roles in homeostasis, angiogenesis, neurogenesis, and innate immunity.

Haley Axelrod et al 2014 <https://www.ncbi.nlm.nih.gov/pmc/articles/PMC4253401/>

Martha Wium et al Cancers 2021, 13, 1521. <https://doi.org/10.3390/cancers13071521>

Response: Thanks for your reminding. We discussed the important roles of AXL in non-cancer cells in introduction section: “Although recent studies demonstrated that AXL may also play an important role in noncancer cells, including neurons, endothelial cells, and immune cells, with roles in homeostasis, angiogenesis, neurogenesis, and innate immunity that may raise concern for potential haematologic and/or immune side effects” (**line 10-13, page 5**). We also cite two references as you mentioned (**ref 17-18**).

3- potential issue regarding the absence of AXL in normal tissues (tumor specificity): poor quality images, poor resolution, renders very difficult proper interpretation of the data Figure 1. Therefore, tumor specificity of AXL remains questionable. Figure 1: shows negative staining in a panel of normal tissues. I don't see evidence that AXL is not expressed in “non-tumour” tissues. Current Microphotographs (or table S1) are not convincing and do not provide sufficient resolution to appreciate if AXL is expressed by tumor cells and not by normal cells (and which cell types), tissues. Clarification is required on this aspect. As mentioned above, It is important in the context of the literature establishing that AXL can be expressed in multiple cell types like endothelial cells, and myeloid cells including macrophages as well as lymphocytes such as NK cells.

To be more convincing, instead of immunoblots, it would be nice to see immunostains with cancerous and adjacent non-tumor lung tissues (non-malignant epithelium, areas with vascular cells, immune infiltrates). increase magnification and quality for normal tissues, or provide additional images as supplementaries.

Response: We appreciate the very helpful advice. In order to show tumour specificity

of AXL, we have re-captured the microphotographs and increased magnification and quality for NSCLC tissues in **figure 1 (line 9, page 7)**. We provided the additional AXL positive primary lung tumour tissues in **figure 1a**.

Fig. 1a Detection of AXL expression in NSCLC samples by IHC. Representative images (magnification $\times 400$) are shown. Scale bar represents 100 μm .

Meanwhile, we captured images with cancerous and adjacent areas with immune infiltrates, vascular cells, or adjacent non-malignant tissues. We have inquired pathologists in our hospital and reveal that AXL expression was not found in immune infiltrates (**black arrow**), vascular cells (**blue arrow**) or adjacent non-malignant tissues (**a**), but in lung cancer tissues (**red arrow**) in **figure S2 (line 1, page 49)**.

Fig. S2 AXL expression in human cancerous but not in adjacent non-tumour lung tissues. AXL detection in NSCLC cancerous (**b/c, red arrow**) and adjacent non-tumour lung tissues, including areas with non-malignant tissues (**a**), vascular cells (**b, blue arrow**) and immune infiltrates (**b/c, black arrow**). Representative staining images (magnification $\times 400$) are shown. Scale bars represent 100 μm . Data summarize one independent experiments.

Moreover, with staining in a panel of normal tissues, we did not see obvious AXL expression in twenty-two types of normal tissues in **figure S1 (line 1, page 48)**.

Fig. S1 AXL expression in human normal tissues. Twenty-two different human normal tissue samples were immunostained with an anti-AXL antibody to determine AXL expression level. Representative staining images (magnification $\times 400$) are shown. Scale bars represent 100 μm . Paraffin-embedded A549 and HCC827 cells served as positive and negative control, respectively. Data summarize one independent experiments.

4- The choice of the CAR generation is unclear

It appears that authors have used 3rd generation CAR-T (CD28+41BB+CD3 endodomains) that are not entered in clinical practice. Why? 2nd generation CAR-T (2 endodomains) are approved in certain indications and known to be relatively safe in patients. Less is known about 3rd generation CAR-T cells.

For sake of clarity, authors should explain this choice, and whenever relevant, additionally include a comparison with 2nd generation CAR-T cells. For instance, a significant improvement of the significance would be to show that the combinatory MWA/CART approach has significant effects when integrating 3rd generation CAR-T, but more limited effects with 2nd AXL-CART cells.

Response: Thanks for the critique. 2nd generation CAR-T constructs have been approved in certain indications. We also agree that 2nd generations are well-believed to be relatively safe in patients.

Many studies indicated that 2nd generation CAR-T cells may be limited *in vivo* in certain solid tumour. Some researchers compared several 2nd generation and 3rd generation CARs targeting different tumour antigens, including mesothelin, PSMA and glypican 3, for antitumour efficacy in different tumour models. Indeed, 3rd generation CARs exhibited higher efficacy in causing tumour regression in certain tumour types. (Leukemia. 2017. doi: 10.1038/leu.2017.249; Molecular Therapy. doi: 10.1038/mt.2009.210; Clin Cancer Res . doi:10.1158/1078-0432.CCR-18-2559).

In our previously results, we also observed that 3rd generation CAR T target mesothelin showed strong anti-tumour activity against gastric cancer both *in vitro* and *in vivo* (Journal of Hematology & Oncology. doi.org/10.1186/s13045-019-0704-y).

In our preliminary study, we attempt to compare MWA + 3rd generation AXL-CAR T and MWA + 2nd generation AXL-CAR-T. We found that MWA combined with 3rd generation CAR T cells potentially resulted in superior tumour growth restriction compared to any of the second generation CAR designs + MWA, **as shown in figure below.**

The above studies suggested that the 3rd generation CAR T cells may be potent in anti tumour efficacy. Therefore, we selected 3rd generation CAR T cells containing CD28,

4-1BB, and CD3 ζ cytoplasmic domains for treatment of NSCLC in our further studies.

Figure. Combination of MWA and 3rd or 2nd generation AXL-CAR-T cells against A549-bearing NSG mice. NSG mice received a subcutaneous injection of 2×10^6 A549 cells. When tumour volume reached about 200 mm^3 , MWA combined with 3rd or 2nd generation AXL-CAR-T cells were administrated to treat A549-bearing tumour. Tumour volume was measured every three days. Data summarize one independent experiment and presented as the mean \pm SD. * $P < 0.05$, ** $P < 0.01$, *** $P < 0.001$.

5- A technical concern, the number of CAR-T infused in mice

The number of CART cells injected in mice is quite high (1×10^7). It is common to see 1 to 10×10^6 CAR-T injection in preclinical studies, and in comparison to patients typically receiving $1-10 \times 10^6$ per kg of bodyweight. Authors should clarify this point and if they have dose response data, they should include it.

Response: We appreciate the comment. The number of CAR-T cells infused in mice here was firstly based on a previous published study (Claudin18.2-Specific Chimeric Antigen Receptor Engineered T Cells for the Treatment of Gastric Cancer. JNCI, 2019. doi:10.1093/jnci/djy134.). In their study, one dose of 1×10^7 Claudin18.2-specific CAR T cells was intravenously injected via the tail vein in $200 \mu\text{L}$ of phosphate-buffered saline by Jiang et. al. to kill gastric cancer, and they achieved satisfactory anti-tumour results and good safety profile *in vivo*.

Secondly, we did perform experiments to test the dose-response efficacy at different

CAR T cell numbers. Generally, 2.5×10^6 , 5×10^6 or 1×10^7 AXL-CAR T cells was intravenously administrated via the tail vein against A549-bearing NSG mice, as shown in **figure below**. With dose increasing, better tumour suppression was actually achieved. We agree that the number of CAR-T cells infused here is important to suppress tumor growth. Therefore, the optimal doses need to be investigated in further clinical studies.

Figure. Dose-response killing efficacy of AXL-CAR T cells against A549 CDX models. NSG mice received a subcutaneous injection of 2×10^6 A549 cells. When tumour volume reached about 200 mm^3 , 2.5×10^6 , 5×10^6 or 1×10^7 AXL-CAR T cells were intravenously administrated via the tail vein to treat A549-bearing tumour. Tumour volume was measured every three days. Data summarize one independent experiment and presented as the mean \pm SD. * $P < 0.05$, ** $P < 0.01$, *** $P < 0.001$.

6-Some additional analysis would be important to support the gain of adding MWA to CAR-T therapy:

-in light of their claim that MWA therapy could potentiate CAR-T effects. It would be keen for the authors to address the question whether MWA could help reduce the dose of CAR-T cells infused, thus limiting potential side effects and toxicity.

Response: Thanks for the very helpful comment. In previous preliminarily study, we noticed that MWA could help reduce the dose of CAR-T cells in combination group.

In revised figure 3, even in MWA+ 10^7 AXL-CAR T cells group, the highest dose

used in current study, no obvious deleterious effect on the important normal organs was observed in the NSG mice (**figure 3i, line 1, page 13**). Therefore, the combination strategy used here may be safe and rational.

Fig. 3i Histopathological analysis of murine organ tissues by haematoxylin and eosin (H&E) staining. Representative staining image fields (magnification $\times 400$) are shown. Each scale bar represents 50 μm .

- Figure 3; as A549 seems to be a good model in Fig2 experiments. I was surprised that authors did not include A549 growth curves using MWA approach combined with CAR-T in Figure 3

Response: Thanks for the mention. We added A549 growth curves using MWA approach combined with CAR-T in **figure 3c (line 1, page 13)**.

Fig. 3c Tumour volume of mice subcutaneously injected with A549 cells using MWA approach combined with CAR-T.

-Figure 3; some control conditions with “CD-19 CAR-T +/- MWA” are missing to check tumor antigen specificity of the MWA effects. I wonder if MWA could also potentiate off-targets/side/toxicity effects.

Response: Thanks for your suggestion. To check tumour antigen specificity of the MWA effects, “CD-19 CAR-T +/- MWA” as the control was added in **figure 3**.

Our data in **figure 3i** show that no obvious off-targets/side/toxicity effects are observed in either MWA alone or in combination group (**line 1, page 13**).

Fig. 3b/c Tumour volume of mice subcutaneously injected with HCC827-ER3 and A549 cells using MWA approach combined with CAR-T.

-another interesting experiment would be to use AXL negative HCC827 xenograft to test the combination strategy.

Response: Thanks for the suggestion. We added the AXL negative HCC827 xenograft to test the combination strategy. As shown in **figure S6**, AXL-CAR T alone or combination strategy did not inhibit tumour growth (**line 1, page 54**).

Fig. S6 Combination of MWA and CAR T cells against HCC827-derived tumour. NSG mice received a subcutaneous injection of 2×10^6 HCC827 cells. When tumour volume reached about 200 mm^3 , mock, MWA +/- 1×10^7 CD19- or AXL-CAR T cells were administrated. Tumour volume was measured every three days (a). Tumour weight was calculated at the end point (b). Data summarize one independent experiment and presented as the mean \pm SD. * $P < 0.05$, ** $P < 0.01$, *** $P < 0.001$.

7-for sake of clarity and in the interpretation of analysis, improvement is required Figure 4 and Figure 5: bad quality IHC stainings, luminosity. It would be great if one could discriminate lymphocytes from tumor cells. what type of HLA is stained (HLA ? A, B, C ?) Authors should correct figure labels.

Response: Thanks for the critique. We provided the improved photos in **figure 4 (line 1, page 16)** and **figure 5 (line 1, page 19)**.

As shown in **figure 4d**, the infiltrated CAR T cells were stained with GFP protein and can be easily discriminated by immunofluorescence (green marker).

Fig. 4d Representative immunofluorescence staining images of GFP-marked CAR-T cell infiltration into cancer tissues from the NSCLC PDX bearing mice. Formalin-fixed, paraffin-embedded mouse tissue sections were stained for GFP (green) and DAPI (blue). Representative staining image fields (magnification $\times 200$) are shown. Scale bar, $100 \mu\text{m}$.

In **figure 4b**, HLA-A was used in this study. We have corrected figure labels accordingly.

8-(Figure 7A) Because the HIF-alpha has a very short-half time (approx. 5 mins), and because WB experiments require a certain number of cells, numerous investigators have reported difficulties in detecting HIF-alpha from purified lymphocytes. I wonder how authors generated such data. It's not clear if CAR-T cells were purified from the tumor bulk (from how many mice) before protein extraction? HIF-alpha expression could come from contaminating tumor fraction. More evidence is necessary to associate HIF deregulation in CAR-T and MWA.

I would suggest to attenuate statement, remove this data, or provide additional evidence that HIF/or HIF pathway is deregulated upon MWA treatment in CAR-T.

Response: Thanks for your suggestion. More evidence is necessary to associate HIF-alpha deregulation in CAR-T and MWA in the future work. We agree and decided to remove this data.

9- clarification required on the impact of MWA on TME and CART.

The reported mechanisms of the effect of MWA are still poorly defined and the cellular components involved unclear. For instance, it's not clear which CART subpopulations express the different markers (CD4+,CD8+, or regulatory-like CAR-T subsets?) Figure 6 and 7. It seems important yet to clarify this point, especially because the proposed relation with hypoxia is hard to interpret, with data at odds with recent reports.

Indeed, the authors made the assumption that hypoxia has deleterious effects on T cells, which contrast with recent reports that consistently showed improvement of CAR-T and T cells effector functions upon hypoxic stress. In fact, there is a possibility that the MWA effects are independent of hypoxia or involve complex metabolic loops. Thus, some 'toning down' of the conclusions on this aspect of the manuscript would be important, or additional evidence required in order to draw firm conclusions."

Response: Thanks for the advices.

In figure 6a, the analysed cells were from CD8+ CAR T cells.

In figure 6c/d/e/g and figure 7, the analysed cells were from CD3+ CAR T cells.

In figure 6f, the analysed cells were from CD4+ CAR T cells.

Indeed, it is hard to clearly elucidate the mechanism of MWA effects on TME at current stage because of TME complication. We agree that the MWA effects are independent of hypoxia or involve complex metabolic loops. It is necessary to investigate this area through comprehensive analysis in the future work.

Therefore, to be more prudent and rational, we deleted the claim “by alleviating hypoxia” in **line 1-2, page 23**, and “toned down” this claim in results section in **line 7-9, page 24**.

10-Unfortunately, authors do not discuss the literature on this aspect and they should add relevant references like...

<https://doi.org/10.1172/JCI85834>

VHL-deficient CD8+ CAR T cells accumulate in tumors with an enhanced Trm-like phenotype. J Clin Invest. 2021;131(7):e143729 <https://doi.org/10.1172/JCI143729>

<https://doi.org/10.1016/j.ccell.2017.10.003>

<http://dx.doi.org/10.1016/j.celrep.2017.08.071>

Response: Thanks for the suggestion. We discussed and cited these papers in discussion section: “Previous studies have suggested that pre-processed hypoxic CTLs or CAR T cells exhibit better antitumor ability. Our results seem contrary to these findings because MWA induced relatively higher PO₂ levels. However, it is reasonable and consistent when considering that either mono- or comb-CAR T cells were maintained in hypoxic conditions. That is, we did not reverse hypoxia but alleviated hypoxia in the NSCLC PDX tumours with MWA. This change may be beneficial for CAR T cells to perform their function. Unfortunately, the mechanism underlying the enhanced comb-CAR T cell function induced by PO₂ fluctuation (and which O₂% may be better) is still unknown. We suppose that PO₂ fluctuation is not an independent factor that changes comb-CAR T cell properties, it might be the complex TME modifications induced by MWA that potentiate the comb-CAR T performance and ultimately result in better tumour regression. Clearly, this is an important area requiring future detailed, comprehensive analysis.” (**line 9-22, page 29**). We also cite

four references as you mentioned (ref 41-44).

11- related to my previous remark, Figure 6, CD25 is typically used as an activating mark of Tregs and could be a mark of Treg-like conversion of CAR-T. A major concern in the field. MWA seems to promote OxPhos metabolism and Treg are known to preferentially use OxPhos metabolism. it would be judicious to additionally include Foxp3 analysis and discriminate CD8+ and CD4+ populations. CD69 is a well-known marker of T cell activation, resident memory phenotype, also reported as an hypoxia-responsive gene. I would also suggest to analyze this marker.

Response: Thanks for the helpful suggestion.

It has been reported that the quiescent T cells (e.g., naive and memory T cells), like most cells in normal tissues, use OXPHOS to meet energy demands ([https://doi.org/10.1016/S1074-7613\(01\)00205-9](https://doi.org/10.1016/S1074-7613(01)00205-9); [10.1016/j.immuni.2011.12.007](https://doi.org/10.1016/j.immuni.2011.12.007)). Our results are consistent with the previous study as we observed higher memory phenotype in comb-CAR T than mono-CAR T. Thus, it is reasonable that OxPhos metabolism was detected higher in comb-CAR T cells.

Moreover, CD25 was also used as activation marker of CAR T cells by Watanabe et. al. (doi.org/10.1172/jci.insight.99573). and Wang et. al. (doi.org/10.1038/s41467-020-20696-x). Our results are consistent with these two studies.

To be more convincing, CD4/CD25/Foxp3 and CD69 are also analysed in revision. We found the CD4⁺/CD25⁺/Foxp3⁺ marker was at low level in both comb- and mono-CAR T cells (**figure 6f, line 1, page 22**). And CD69 was expressed higher in comb-CAR T than mono-CAR T, as shown in **figure 6a (line 1, page 22)**.

12- Figure 5h (Po2 measures): except at day 1, I don't see much difference. Is this significant ? As a side note, for non-expert, it's difficult to appreciate what pO2 level one should consider hypoxic or normoxic. Discussion and/or text could be expanded to include a bit more on this point.

Response: Thanks for the suggestion. In **figure 5g**, PO₂ in combination group was significantly higher compared with AXL-CAR T and Mock groups ($P < 0.001$).

According to previous study, the majority PO₂ in tumour less than 20 mmHg, which means hypoxic, and less than 5 mmHg means extremely hypoxia. Approximately, 50 mmHg means normoxic (doi.org/10.1073/pnas.042671399; [10.1089/ars.2013.5378](https://doi.org/10.1089/ars.2013.5378)).

We have added description in discussion section (**line 12-14, page 29; ref 45-46**).

13-The authors should improve discussion of the preclinical data and put them in context of the current knowledge and treatment strategies:

-potential benefit of their approach compared to typical combination therapy and what benefit compared to other AXL-targeting approaches. There are only 31 References and only 4 papers on AXL including their paper (14-17). No recent review or work discussing other AXL targeting strategies already used in clinics, or in clinical development. there are numerous recent reviews covering this aspect that authors could use. (see Tanaka, M. et al <https://doi.org/10.3390/ijms22189953> for update).

Response: Thanks for the advice. We improved the discussion to typical combination therapy in discussion section: “Several Phase I/II studies are underway to evaluate Type I/II AXL kinase inhibitors in combination with chemotherapy or immune checkpoint inhibitors in various solid tumours, however, the overall antitumour efficacy are far from satisfaction. Our data were consistent with the reports that photothermal therapy or cytokine-armed oncolytic adenoviruses facilitated the accumulation and effector function of CAR T cells within solid tumours. Moreover, our combination strategy may show advantages to photothermal combination therapy for deep tumours because of the good penetration and accessibility of MWA needles. What's more, considering that MWA is a frequently-used method in lung cancer, suggesting that our combination strategy has high translational promise in clinical practice.” (**line 2-12, page 27**). We cited the references as you mentioned (**ref 28**).

And to other AXL-targeting approaches in introduction: “Although recent studies demonstrated that AXL may also play an important role in noncancer cells, including neurons, endothelial cells, and immune cells, with roles in homeostasis, angiogenesis, neurogenesis, and innate immunity that may raise concern for potential haematologic

and/or immune side effects, a series of AXL-targeted small-molecule inhibitors have been investigated in preclinical and clinical trials, including R428 and BPI-9016M, which showed favourable safety and pharmacokinetic profiles in NSCLC patients. Moreover, a high number of AXL-targeted monoclonal antibodies, such as hMAb173 for renal cell carcinoma, D9 and E8 for pancreatic cancer, and YW327.6S2 for NSCLC and breast cancer, have shown anticancer efficacy in preclinical and earlier clinical trial stages. In addition, AXL-targeted CAR T cells showed killing effects in breast cancer models in vivo and various tumour models in vitro. However, those monotherapies exhibited only moderate antitumour efficacy in previous studies, and none of the studies focused on the use of AXL-targeting CAR T cells for the treatment of NSCLC. There is a clear unmet medical need to improve the response of AXL-CAR T cells in NSCLC.” (line 10-22, page 5; line 1-3, page 6).

One question that arises is what would be the benefit of AXL-CAR-T compared to those and to what price in terms of toxicity.

Response: We appreciate the comment. To our knowledge, compared to the aforementioned AXL-targeting approaches, it is the first time to systematically observe the potential “off-target” side effects in our study. We showed a good safety threshold in **figure 3i (line 1, page 13)**, because there was no obvious damage was observed in the organs from mice treated with MWA and/or AXL-CAR T cells by HE staining. Thus, AXL-CAR-T might be safe at least in animal study.

Moreover, we compared the group AXL-CAR T and AXL inhibitor (R428) to reflect the advantage of CAR-T that with potential side effects for A549-bearing mice. **As shown in figure below**, no obvious damages were observed in both AXL-CAR T and AXL inhibitor group.

However, individualized detection of AXL expression is still needed when considering clinical application of AXL-CAR T cells.

Figure. Safety and anti-tumour efficacy of AXL-CAR T cells and R428 against A549-derived tumour. NSG mice received a subcutaneous injection of 2×10^6 A549 cells. When tumour volume reached about 200 mm^3 , mock, R428 (50 mg/kg, po, bid) or AXL-CAR T cells (10^7 , i.v.) were administrated. (a) Tumour volume was measured every three days. (b) Tumour weight was calculated at the end point. (c) Histopathological analysis of murine organ tissues by haematoxylin and eosin (HE) staining. Representative staining image fields (magnification $\times 400$) are shown. Each scale bar represents $50 \mu\text{m}$. Data summarize one independent experiment and presented as the mean \pm SD. * $P < 0.05$, ** $P < 0.01$, *** $P < 0.001$.

-It's not clear for me why their method integrating AXL targeting would do better than any other CAR-T cells directed to solid tumors. (Niels Schaft <https://www.mdpi.com/2072-6694/12/9/2567>)

Response: Thanks for the critique. We confirmed that AXL-CAR T cells alone only had a moderately inhibitory, but not a completely suppressive effect against AXL-positive NSCLC subcutaneous and pulmonary metastatic tumours *in vivo* (figure 2, line 1, page 10). This is consistent to previous study (Niels Schaft, <https://www.mdpi.com/2072-6694/12/9/2567>). The enhanced *in vivo* anti-tumour efficacy in figure 3/4 may due to the use of combination strategy. However, without comprehensive studies, we cannot conclude the method integrating AXL targeting is better than any other CAR-T cells directed to solid tumors.

Reviewer #3 (Remarks to the Author): with expertise in CAR-T, lung cancer

This is a nice study which demonstrates that microwave ablation (MWA) can synergize with CAR-T cells to improve tumor control in xenograft models of NSCLC. The authors identify AXL as a potential CAR target that is expressed in a subset of NSCLC and shows limited expression in normal tissues, and develop AXL-specific CAR-T cells that show moderate activity in xenograft models of AXL+ NSCLC. MWA is shown to synergistically enhance tumor control by AXL CAR-T cells in a patient-derived xenograft model. Mechanistically, MWA is shown to increase blood flow, vascularization, and pO₂ levels in tumors. Enhanced tumor control in the combination therapy group is associated with increased frequency of CAR-T cells in tumors, which show reduced HIF-1 α expression, increased oxidative metabolism, and a more “central memory” phenotype.

Overall, the effect of MWA on CAR-T cell activity and tumor control is impressive and suggests that this is a promising combination therapy to pursue in the clinic. However, the authors make several claims about the mechanism of synergy between MWA and CAR-T cells that are not appropriately supported by the data. These claims either require more data to support, or should be modified:

- 1) The authors claim that “MWA induced the activation and memory phenotype, as well as reduced exhaustion of tumor-infiltrating AXL-CAR T cells in NSCLC PDX tumors.” It is not clear from the data that the functionality of tumor-infiltrating CAR T cells is changed by MWA. It is possible that enhanced tumor control is mediated primarily by increasing CAR T cell infiltration into tumors via enhanced vascularization/blood flow/etc rather than due to qualitative changes in CAR T cell function. MWA is shown to increase total cytokine levels in the tumor and blood, but this could simply be a product of the increased frequency of CAR T cells observed in the tumor and blood. Whether CAR T cells are more functional on a per-cell basis requires analysis of cytokine production by CAR T cells, either through intracellular cytokine staining or by sorting CAR T cells and examining cytokine production by

ELISA/Luminex after restimulation ex vivo.

Response: Thanks for the comments. We agree that enhanced tumour control may be mediated primarily by increasing CAR T cell infiltration into tumours via enhanced vascularization/blood flow/etc rather than due to qualitative changes in CAR T cell function. We have changed our claim “MWA induced the activation and memory phenotype, as well as reduced exhaustion of tumour-infiltrating AXL-CAR T cells in NSCLC PDX tumours” to “Together, these data indicated that the comb-CAR T cells displayed higher cytokines release, more activation/memory differentiation and less exhaustion than mono-CAR T cells. Although it is not clear from these data that the functionality of the tumour-infiltrating CAR T cells was directly changed by the MWA, we believe that enhanced tumour control may have been mediated by either improvement of function or increased CAR T cell infiltration into tumours induced by MWA effects.” (line 7-13, page 21).

2) The authors claim that the “oxidative features in comb-CAR T cells most likely supported central memory differentiation and T cell persistence.” Although increased pO₂ levels in the tumor, increased oxidative metabolism in CAR T cells, and increased memory marker expression by CAR T cells are all observed, no data is provided to support a functional link between these phenotypes. Central memory T cells are known to exhibit greater oxidative metabolism, but whether changes in metabolism drive memory differentiation, or whether memory differentiation drives oxidative metabolism is not shown. Additionally, tumor burden/antigen load is known to strongly influence T cell differentiation and drive exhaustion, so it is possible that the enhanced memory phenotype of CAR T cells in MWA-treated tumors is simply due to the dramatically smaller tumors in these mice at this time point, and that metabolism of CAR T cells is altered simply as a result of memory differentiation.

Response: We appreciate and pretty much agree with the comments. This claim might be unthoughtful. We attenuated this statement and removed this claim. A functional link between these phenotypes needs to be comprehensively investigated in the future study. The final conclusions are not affected in the manuscript.

3) The authors claim that “the combination therapy increased mitochondrial oxidative metabolism and decreased HIF-1a expression in tumor-infiltrating CAR T cells by alleviating TME hypoxia.” Although MWA is shown to increase tumor pO₂ levels and CAR-T cells show increased oxidative metabolism, no data is provided to support a functional link between these two observations. It is not shown whether enhanced oxidative metabolism is due to decreased hypoxia in the tumor, or whether it results indirectly from changes in tumor burden that preferentially induce memory differentiation of CAR T cells.

It is not necessary to provide data demonstrating a functional link between all these readouts, but these claims should be tempered to account for the fact that they are correlative.

Response: Thanks a lot and we agree with your advice. The claim “the combination therapy increased mitochondrial oxidative metabolism and decreased HIF-1a expression in tumour-infiltrating CAR T cells by alleviating TME hypoxia.” should be attenuated. We changed this claim to “The combination therapy increased the mitochondrial oxidative metabolism of tumour-infiltrating CAR T cells” (**line 13-15, page 2**).

Reviewer #4 (Remarks to the Author): with expertise in MWA

This manuscript reported that AXL-CART combined with MWA shows superior anti-tumor effect depending on a modification of TME. Overall, the work is of interest but there are many areas that need to be addressed. The major issues are listed below:

Introduction

1. a series of AXL-targeted small-molecule inhibitors have been investigated in clinic with favourable safety and pharmacokinetic profiles in NSCLC, why target AXL with CAR-T cells for the treatment of NSCLC should be explained, what’s the potential advantage of AXL-CAR-T to R428 and BPI-9016M when combined with MWA.

Response: Thanks so much for the valuable comments. We have addressed the potential advantage of AXL-CAR T to R428 and BPI-9016M when combined with

MWA: “In light of the potential immune-boosting capability of MWA and the hypofunction of CAR T cells in solid tumours, the combination of MWA and AXL-CAR T cells may be an attractive approach against NSCLC” (line 3-4, page 6). And in discussion section “First, the proinflammatory chemokines and cytokines released from the ablated tissue or tumour cells, as well as the disruption of local extracellular matrix and tissue components, provided a local inflammatory response, which might have generated specific immunity, promoted CAR T cell activity and eventually enhanced CAR T cell infiltration. Second, tumour antigens released from tumour cells destroyed by MWA could also have enhanced the killing potency of CAR T cells because early and direct antigen exposure benefits the activation and lytic capability of CAR T cells” (line 5-12, page 28)

2. The combinational value of MWA with CAR-T should be discussed, why achieved $1+1>2$?

Response: Thanks for the comment. Actually, we have discussed this part in discussion section. The $1+1>2$ effect may be explained that MWA created a relatively permissive environment in tumour bearing mice, which enhance the activation, accumulation, persistence, and antitumour efficacy of tumour-infiltrating CAR T cells *in vivo* (line 16-19, page 28).

Methods

1. What’s the sequence of MWA and CAR-T when combination therapy?

Response: Thanks for the mention. AXL-CAR T cells were intravenously injected after MWA. We added this information in Materials and Methods (line 14, page 34).

2. In the sixth part (page 21-22 MWA induces activation and memory phenotype of tumour-infiltrating AXL-CAR T cells and reduces their exhaustion in PDX tumours), the author showed that on days 42 ,when CAR-T cells had higher infiltration, they expressed more active markers. I think the same study should be carried out on the days 33 (3 days after treatment). To confirm whether AXL-CAR-T can change with

time like HA or IFP.

Response: Thanks for the suggestion and we added this data on day 33, as shown in **figure 6a (line 1, page 22)**. Compared to day 42, more CD69 and less CD95 expression were observed in tumour-infiltrating AXL-CAR T cells on day 33.

3. Minor problem might be informal acronym, such as CDX and PDX, which were not explained before.

Response: Thanks for your suggestion. CDX and PDX represent cell-derived xenografts (**line 16-17, page 12**) and patient-derived xenografts (**line 15, page 14**), respectively. We defined the full names of two acronyms in the manuscript.

4. Study should add and compare the group AXL-CAR-T and AXL inhibitor to reflect the advantage of CAR-T that with potential side effects for solid tumor.

Response: Thanks for your advice. We performed this study as suggested. **As shown in figure below**, no obvious damages were observed in both AXL-CAR-T and AXL inhibitor group.

Figure. Safety and anti-tumour efficacy of AXL-CAR T cells and R428 against A549-derived tumour. NSG mice received a subcutaneous injection of 2×10^6 A549 cells. When tumour volume reached about 200 mm^3 , mock, R428 (50 mg/kg, po, bid) or AXL-CAR T cells (10^7 , i.v.) were administrated. **(a)** Tumour volume was measured every three days. **(b)** Tumour weight was

calculated at the end point. (c) Histopathological analysis of murine organ tissues by haematoxylin and eosin (H&E) staining. Representative staining image fields (magnification $\times 400$) are shown. Each scale bar represents 50 μm . Data summarize one independent experiment and presented as the mean \pm SD. * $P < 0.05$, ** $P < 0.01$, *** $P < 0.001$.

5. Does locoregional treatment will affect the expression of AXL in NSCLC? In other words, whether comb-CART treatment has the similar anti-tumor effects in AXL-negative NSCLC?

Response: We appreciate the question. We added the AXL negative HCC827 xenograft to test the anti-tumour effects of combination strategy. As shown in **figure S6 (line 1, page 54)**, CAR T cells alone or combination strategy did not suppress HCC827-derived subcutaneous tumour.

Fig. S6 Combination of MWA and CAR T cells against HCC827-derived CDX tumour. NSG mice received a subcutaneous injection of 2×10^6 HCC827 cells. When tumour volume reached about 200 mm^3 , mock, MWA +/- 1×10^7 CD19- or AXL-CAR T cells were intravenously administered via the tail vein against HCC827-bearing tumour. Tumour volume was measured every three days (a). Tumour weight was calculated at the end point (b). Data summarize one independent experiment and presented as the mean \pm SD. * $P < 0.05$, ** $P < 0.01$, *** $P < 0.001$.

Results

1. In fig 4d, to show the infiltration of AXL-CART in the tumor tissue, however, it is not reasonable for indication of AXL-CART by using CD3 antibody (a T cell marker),

a specific AXL antibody co-stain with CD8 was needed. MWA can induce inflammatory reaction, so the elevated CD3 may not be induced by AXL-CART.

Response: Thanks for your advice. Because the NSG mice lack of T cell, so the anti-human CD3 antibody staining in **figure 4d (line 1, page 16)** represents the T or CAR T cell infiltration. However, all the AXL-CAR T cells carry a reporter gene-*GFP*, we therefore added immunofluorescence to detect GFP (only AXL-CAR T cell contain GFP). This data is enough to prove the AXL-CAR T cell infiltration in tumour.

Fig. 4d Representative immunostaining images of GFP CAR-T cell infiltration into cancer tissues from the NSCLC PDX bearing mice. Formalin-fixed, paraffin-embedded mouse tissue sections were stained for GFP (green). Representative staining image fields (magnification $\times 200$) are shown. Each scale bar represents 100 μm .

2. In fig 4e, we notice that the number of CART was increased both in the PBMC and tumor tissue. However, the total cell mass of mono-CART group was twice more than it in the comb-CART group. Moreover, in their previous results, there are more AXL-CARTs were accumulated in the tumor region under comb-CART group. What was the reason for the expanding AXL-CART population? Proliferation or cell death resistant?

Response: Thanks for your thoughtful questions. Indeed, there are more AXL-CARTs were accumulated in the tumour region under comb-CART group. The detailed

mechanism maybe complicated. But one phenomenon (**figure 6g, line 1, page 22**) that observed is that exhaustion markers, such as PD1, TIM3 and CTLA4, were expressed lower in comb-CART group than that in mono-CART group. Thus, we suppose that the cell death resistant may be primary reason that why the number of CART was increased both in the PBMC and tumour tissue in comb-CART group (**figure 4e/f, line 1, page 16**).

Another possible reason is that the MWA in combination group created a relatively permissive environment in tumour bearing mice, which enhance the accumulation of tumour-infiltrating CAR T cells *in vivo* (**line 16-19, page 28**).

3. Continue to question 3, although they provide sufficient evidence for supporting the hyperfunction of CART combined with MWA, given the cell amounts was more than twice in comb-CART (figure 4e), does the main reason that contribute to the upregulation of cytokines in combined group is associated with a functional enhanced AXL-CART or just due to the expanding AXL-CART population?

Response: We appreciate the comment. Based on our data in **figure 4 (line 1, page 16) and figure 6 (line 1, page 22)**, it is possible that the upregulation of cytokines in combined group is mediated primarily by increased CAR T cell infiltration into tumours.

4. In Supplementary fig 4, the remaining tumor volume reduced sharply under the condition of 10W and 40s/10w and 80s, why don't choose a moderate parameter (10W and 20s)?

Response: We appreciate the question. There are several reasons that we selected 10 W and 40s.

1. In our previous experiment, we had compared 10W/20s + AXL-CAR T and 10W/40s + AXL-CAR T in terms of anti-tumour efficacy. We found that 10 W, 40s + AXL-CAR T yielded better tumour control (**figure below**).

2. The average residue tumour volume in 10 W/20s and 10W/40s have no statistical difference (**figure S5, page 53**).

3. In order to mimic tumour residue after clinical ablation, we selected a suitable ablation parameter to ablate tumour moderately, neither excessive or mild. Thus, we select 10W/40s as ablation parameter.

Figure. Combination of CAR T cells with different condition (10W/20s, 10W/40s) MWA against HCC827-ER3-derived CDX tumour. NSG mice received a subcutaneous injection of 2×10^6 HCC827-ER3 cells. When tumour volume reached about 200 mm^3 , mock, 10^7 AXL-CAR T cells +/- MWA were administrated. Tumour volume was measured every three days. Data summarize one independent experiment and presented as the mean \pm SD. * $P < 0.05$, ** $P < 0.01$, *** $P < 0.001$.

Discussion

1. In the last part of this work, they discuss that MWA can alleviate hypoxia in CART, which result in a decreased expression of T cell exhaustion markers. Does knockdown/knockout of *Hif1 α* can reverse the phenotype of mono-CART?

Response: Thanks for the suggestion. Because of the limited life-span of sorted CARTs, it is difficult to knockdown/knockout *Hif1 α* gene directly in CARTs.

However, a previous *in vitro* study supports that *HIF1 α* -deficient tumour-infiltrating T lymphocytes were characterized by a drop in expression of the the exhaustion markers PD-1, LAG3, and TIM3 (<https://doi.org/10.1016/j.ccell.2017.10.003>). This may help us to clarify that knockdown/knockout of *Hif1 α* can reverse the phenotype of mono-CART.

2. In discussion, they explore the limitation of AXL-CART for potential “on-target/off-tumour” effects, but it had been reported that the expression of AXL was ubiquitous (the protein level and mRNA level was shown in the below figures). In this paper, the AXL was mainly exist in the NSCLC, and hardly be detected in the normal tissues.

Response: Thanks for the comments. Although no figure was attached below, we guess you may ask us whether AXL is a specific and safe target for CAR T immunotherapy in NSCLC.

Untill now, there is no 100% tumour-specific expression, or truly perfect targets for CAR T immunotherapy in solid tumours, the majority of them are from tumour-associated antigens (TAA), including widely reported CAR-T targets mesothelin, GPC3, MUC1, DLL3, HER-2, EGFR, CEA, Claudin-18.2, Claudin-6, AFP, B7-H3 and so on (<https://www.ncbi.nlm.nih.gov/pmc/articles/PMC7563774/>). Which means TAAs may also expressed in normal tissues, and have the potential for “on-target off-tumour” side effects. This is the dilemma of current CAR T cell immunotherapy in solid tumours.

Despite these challenges, there are many clinical trails are recruiting patients for above antigen-specific CAR T cell immunotherpey for various solid tumours. Recently, we finished a phase I clinical trail (NCT03198546) targeting GPC3 or Mesothelin against liver cancer or pancreatic cancer in our institution and achieved preliminary progress in safety and efficacy (<https://www.ncbi.nlm.nih.gov/pmc/articles/PMC8323212/>). Therefore, there is still essential and valuable to evaluate new TAAs in different solid tumours.

To our knowledge, in current study, AXL was first used as target for CAR T cell therapy in NSCLC. In **figure 1a (line 9, page 7)**, **figure S1 (line 1, page 48)** and **figure S2 (line 1, page 49)**, AXL expression can be easily found in tumour tissue and raraly found in 22 kinds of human normal tissues, which indicates a good specificity. And our *in vivo* study showed no obvious damage was observed in the organs from mice treated with AXL-CAR T cells by HE staining, which indicated a good safety threshold when considering AXL-CAR T cells can target AXL protein in both human

and mouse tissue (**Fig. 3i, line 1, page 13**).

Moreover, to further evaluate the safety and efficacy of AXL-CAR T cells in advanced AXL-positive lung cancer patients, a phase I clinical trial (NCT03198052) is conducted by our team and currently underway in our institution.

REVIEWER COMMENTS

Reviewer #1 (Remarks to the Author):

The manuscript has been improved significantly, I have appreciated the answers to all the questions, clarified with additional experiments or improvement of the figure quality.

It is very interesting that in figure 3b and 3c the tumour volumes between the group treated with MWA, CD19 CART or AXL-CART are not massive different, instead big difference has showed in term of Metastasis, where AXL-CART reduce drastically the metastasis formation in the treated mice also in not combination with MWA, this could be related to low macrophage infiltration showed in Sup 7, few group have been demonstrated the macrophages migration and tissue infiltration as an important factor to establish the metastatic process.

It could be something interesting to look.

Reviewer #3 (Remarks to the Author):

The authors have sufficiently addressed my concerns.

Reviewer #5 (Remarks to the Author): to replace Reviewer #2

The authors introduced substantial changes to the manuscript to address the reviewers' comments, including new in vivo experiments, and the manuscript is substantially improved. Changes in organization and addition of references and statements describing previous relevant work improve the clarity of the paper. The studies are well-designed and executed and clearly presented. There is a clear need for new therapies for lung cancer and an interest in immunotherapies in particular. There is still concern regarding the impact of the work. The major advance in this manuscript is the combination of CAR T therapy with microwave ablation (MWA) to provide enhanced therapeutic efficacy. Another group has previously demonstrated enhanced CAR T efficacy associated with changes in the TME accompanied by changes in interstitial fluid pressure, increased blood perfusion, and increased recruitment of CAR T cells in response to hyperthermia. Mechanistic observations add some novelty, but are largely correlative and don't address whether the observed changes in the TME are functionally relevant (i.e. causative) for improvements in CAR T efficacy. Presumably this approach could only be applied for local tumor control, since it requires insertion of a probe for MWA and the proposed mechanism involves activation of changes in the local TME to enhance exposure to CAR T cells that wouldn't necessarily be expected to transfer to other tumor sites. It is also not clear from the manuscript how frequently MWA is currently used and under what circumstances. The choice of AXL as a potential CAR T target is interesting with potential to be quite effective given the known impacts of TAM kinase inhibition in the tumor immune microenvironment. However, the true potential of targeting AXL-expressing cells in the context of an intact immune system, where AXL inhibition can reprogram AXL-expressing innate immune cells to activate anti-tumor immunity, is not explored since all of the models used are immune compromised. In addition, given the known roles for AXL in the immune system, it is unclear whether studies in immune compromised mice reflect all potential toxicities associated with this approach.

Other Comments:

The authors should comment on the concordance of their data regarding AXL expression in patient samples with previous publications (e.g. Linger et al, Oncogene, 2013)
Are all of the EGFR-mt samples in Table S2 from EGFR TKI-resistant tumors? If so, the table should be labeled as such to avoid confusion.

In the results section, the figure panel reference for HLA-DR staining should be Figure 3f instead of Figure 3g

Addition of the article “the” in this revised version is often unnecessary and detracts from clarity.

Reviewer 2 point 6 –The authors respond by stating that they have generated data to address the reviewers comment, but don’t show the data. The data should be included in the response to allow the reviewers to evaluate whether the concern has been adequately addressed.

P11 line 23: clarify that HCC827 is an Axl-negative cell line

Need to cite references describing clinical development of AXL antibodies to support the following statement “Moreover, a high number of AXL-targeted monoclonal antibodies, such as hMAb173 for renal cell carcinoma, D9 and E8 for pancreatic cancer²³, and YW327.6S2 for NSCLC and breast cancer, have shown anticancer efficacy in preclinical and earlier clinical trial stages.”

Reviewer #6 (Remarks to the Author): to replace Reviewer #4

This is a very interesting study, with many innovations. But there are a few questions the author needs to answer.

1. What are the noteworthy results?

(1). The data in Fig. 3 are insufficient to show that MWA promotes antitumor activity of AXL-CAR T cells against lung cancer. In this part of the experiment, subcutaneous tumor models were used and 10 W and 40s as parameters sufficient to achieve partial ablation, so the tumor reduction seen in Fig. 3 b/d and Fig. 3 c/e may be caused by physical thermal ablation of MWA rather than enhancing the antitumor effect of AXL-CAR T. Fig. 3 g showed that there was no difference in the number of metastases between AXL-CAR T and MWA + AXL-CAR T groups, and Fig. 3 h showed no difference in lung weight, which also did not indicate that MWA enhanced the anti-tumor effect of AXL-CAR T. Therefore, is there more data to support the conclusion that MWA promotes antitumor activity of AXL-CAR T cells against lung cancer?

(2). Fig7 i Transmission electron microscopy results came from day42 tumor tissue, whether it is the result after preparing a single cell suspension for tumor tissue? Because the background is too clean, unlike the result of directly making ultrathin sections of tumor tissue.

2. Will the work be of significance to the field and related fields? How does it compare to the established literature? If the work is not original, please provide relevant references.

Lung cancer has the highest mortality rate worldwide, with NSCLC accounting for about 85% and 5-year survival rate of only 5%. Improving the treatment status and achieving long-term survival is the most urgent need for patients with advanced NSCLC. In this work, AXL-CAR T combined with MWA was constructed to effectively inhibit the progression of NSCLC, providing an innovative theoretical basis for the treatment of NSCLC. Compared with previous studies, first, this work investigated the role of AXL-CAR T in NSCLC for the first time, and cleverly combined MWA technology to enhance the anti-tumor effect of AXL-CAR T; second, this work also demonstrated that MWA reprogrammed the metabolic pattern of AXL-CAR T while the latter obtained a more activated and memory phenotype; and finally, it was unexpected that this work found that MWA reduced the level of hyaluronic acid, increased higher blood flow rate, elevated partial pressure of O₂, and then remodeled the tumor microenvironment. In summary, the work will be of significance to the field and related fields.

3. Does the work support the conclusions and claims, or is additional evidence needed?

This work requires additional work demonstrating that MWA promotes antitumor activity of AXL-CAR T cells against lung cancer.

4. Are there any flaws in the data analysis, interpretation and conclusions? Do these prohibit publication or require revision?

The data analysis, interpretation and conclusions of this work are not flawed, except for the

conclusion that " MWA promotes antitumour activity of AXL-CAR T cells against lung cancer " which requires more data support.

5. Is the methodology sound? Does the work meet the expected standards in your field?

The methodology in this work is sound. The work meets the expected criteria in lung cancer and MWA field.

6. Is there enough detail provided in the methods for the work to be reproduced?

There is enough detail provided in the methods for the work to be reproduced.

We thank all six reviewers again for their professional comments and advice for our revised manuscript (NCOMMS-21-41470A). We have addressed all the comments raised by the reviewers, as summarized below (Queries in black, response to Reviewers in blue). We request you to consider this article for publication in *Nature Communications*.

REVIEWER COMMENTS

Reviewer #1 (Remarks to the Author):

The manuscript has been improved significantly, I have appreciated the answers to all the questions, clarified with additional experiments or improvement of the figure quality.

It is very interesting that in figure 3b and 3c the tumour volumes between the group treated with MWA, CD19 CART or AXL-CART are not massive different, instead big difference has showed in term of Metastasis, where AXL-CART reduce drastically the metastasis formation in the treated mice also in not combination with MWA, this could be related to low macrophage infiltration showed in Sup 7, few group have been demonstrated the macrophages migration and tissue infiltration as an important factor to establish the metastatic process.

It could be something interesting to look.

Response: Thanks again for your valuable suggestions for our revised manuscript. We would like to address the macrophages migration and tissue infiltration as an important factor to establish the metastatic process in the future study.

Reviewer #3 (Remarks to the Author):

The authors have sufficiently addressed my concerns.

Response: Thank you again for your helpful comments.

Reviewer #5 (Remarks to the Author): to replace Reviewer #2

The authors introduced substantial changes to the manuscript to address the reviewers' comments, including new in vivo experiments, and the manuscript is substantially improved. Changes in organization and addition of references and statements describing previous relevant work improve the clarity of the paper. The studies are well-designed and executed and clearly presented. There is a clear need for new therapies for lung cancer and an interest in immunotherapies in particular.

There is still concern regarding the impact of the work. The major advance in this manuscript is the combination of CAR T therapy with microwave ablation (MWA) to provide enhanced therapeutic efficacy. Another group has previously demonstrated enhanced CAR T efficacy associated with changes in the TME accompanied by changes in interstitial fluid pressure, increased blood perfusion, and increased recruitment of CAR T cells in response to hyperthermia. Mechanistic observations add some novelty, but are largely correlative and don't address whether the observed changes in the TME are functionally relevant (i.e. causative) for improvements in CAR T efficacy. Presumably this approach could only be applied for local tumor control, since it requires insertion of a probe for MWA and the proposed mechanism involves activation of changes in the local TME to enhance exposure to CAR T cells that wouldn't necessarily be expected to transfer to other tumor sites.

Response: Thanks for your helpful comments.

Some studies indicated that MWA could induce abscopal effect, which is the spontaneous regression of a remote no-target tumour (*Rao P. et al. Cardiovasc Intervent Radiol. 2011 Apr;34(2):424-30. Takaki H. et al. Diagn Interv Imaging. 2017 Sep;98(9):651-659.*). Moreover, several preclinical and clinical studies have suggested that MWA can therapeutically induce effective systemic antitumour immune response by the combined immunomodulators (*Chu, K. F. et al. Nat Rev Cancer. 2014 Mar;14(3):199-208.*). Thus, to address whether MWA can enhance antitumour efficacy of CAR T cells in distant site, we established one local (left flank) and distant (right flank) tumour model by inoculating A549 cells in NSG mice. When

the local tumour volume reached approximately 200 mm³, they were allocated to the following groups: A. mock, B. MWA, C. AXL-CAR T (i.v.), D. MWA+AXL-CAR T (i.v.). A549 cells were inoculated in the right flank (distant) at the same day. The tumour growth and CAR T cell infiltration in distant tumour were observed (**Fig. S10, line 15-18/13-21/1-8 and page 12/51/62**, respectively).

We noticed that both MWA alone and AXL-CAR T group displayed mild tumour growth inhibition in the distant (right flank) site. In contrast, MWA+AXL-CAR T group showed not only better distant tumour suppression, but also greater CAR T cell infiltration. (**Fig. S10b/c**).

Although the detailed mechanism of how MWA enhances the anti-tumour effect of AXL-CAR T cells against distant tumour should be further investigated, the previous studies imply that MWA-mediated abscopal effects may enhance activation of the systemic antitumour immune response (*Chu, K. F. et al. Nat Rev Cancer. 2014 Mar;14(3):199-208. Takaki H. et al. Diagn Interv Imaging. 2017 Sep;98(9):651-659.*). Meanwhile, various immune modulating factors such as tumour-associated antigens, danger signals, and cytokines may be released or secreted after MWA treatment (*den Brok M.H. et al, Br J Cancer. 2006 Oct 9;95(7):896-905. Nobuoka D. et al. Int J Oncol. 2012 Jan;40(1):63-70. Schueller G. et al. Int J Oncol. 2004 Mar;24(3):609-13. Erinjeri J.P. et al. J Vasc Interv Radiol. 2013 Aug;24(8):1105-12.*). Tumour-associated antigens released from tumour cells destroyed by MWA may have effect on killing potency of CAR T cells because early and direct antigen exposure benefits the accumulation and function of CAR T cells (*Adusumilli P.S. et al. Sci Transl Med. 2014 Nov 5;6(261):261ra151*). Further, a local inflammatory response induced by MWA may generate specific immunity and effectively impact distant tumour sites (*Sci Transl Med. 2014 Mar 5;6(226):226ra32.*).

Based on previous study and current data, we assume that MWA may promote distant tumour control by increasing accumulation of AXL-CAR T cells through the MWA-mediated abscopal effects.

Fig. S10. Antitumour efficacy of MWA +/- AXL-CAR T cells against A549-derived distant tumour. **a.** Schematic representation of the experiment. Briefly, NSG mice received a subcutaneous injection of 2×10^6 A549 cells in the left flank (local). When tumour volume reached about 200 mm^3 (day 0), mock, MWA +/- AXL-CAR T cells (i.v.) were administrated against A549-bearing tumour. At the same day (day 0), another 2×10^6 A549 cells were inoculated to the right flank (distant). **b.** Tumour volume in the distant flank. **c.** Percentages of CAR T cells (GFP+) in tumour by flow cytometry. * $P < 0.05$, ** $P < 0.01$, *** $P < 0.001$.

It is also not clear from the manuscript how frequently MWA is currently used and under what circumstances.

Response: Based on National Comprehensive Cancer Network (NCCN) guideline, MWA is an important technique in the treatment of inoperable lung cancer (*Ettinger D. S. et al. J Natl Compr Canc Netw. 2021 Mar 2;19(3):254-266.*). And it is frequently used in primary and secondary lung cancer (*Dupuy D.E. et al. Radiology. 2011 Sep;260(3):633-55.*).

The choice of AXL as a potential CAR T target is interesting with potential to be quite effective given the known impacts of TAM kinase inhibition in the tumor immune microenvironment. However, the true potential of targeting AXL-expressing cells in the context of an intact immune system, where AXL inhibition can reprogram AXL-expressing innate immune cells to activate anti-tumor immunity, is not explored since all of the models used are immune compromised. In addition, given the known roles for AXL in the immune system, it is unclear whether studies in immune compromised mice reflect all potential toxicities associated with this approach.

Response: We appreciate the comments. To address the efficacy and safety of the combination strategy, we performed combination therapy in humanized

immunocompetent mice (huNOG-EXL) (**Fig. 8, line 8-22/1-17/1-10/1-3/8-22/1-16/14-16/4-5,10,22/1-2 and page 25/26/27/28/33/34/36/38/39, ref 33/34/49/50, respectively**).

A549 cells were subcutaneously inoculated into the flank of huNOG-EXL mice. When the tumour volume reached about 200 mm³, mice were randomly allocated to four groups: mock (PBS), MWA, AXL-CAR T cells (i.v.), MWA combined with AXL-CAR T cells (i.v.) (**Fig. 8a**). Neither AXL-CAR T cells nor MWA monotherapy was able to suppress tumour growth. Administration of AXL-CAR T cells combined with MWA exhibited effective antitumour response (**Fig. 8b**). Moreover, the tumour-infiltrated T cells were measured. We found that combination treatment resulted in higher donor (GFP⁺) and host T cell infiltration in the tumours than monotherapies (**Fig. 8c/d**). These results highlight the importance of combination treatment in an immunocompetent setting, which possibly activates endogenous adaptive and innate antitumour activity. Macrophages were critical components in innate antitumor immunity. To assess whether combination therapy alters potential macrophage-mediated immune suppression, we analyzed the phenotypes of macrophages in tumours. We found that MWA led to mild elevation of total macrophages (CD68⁺) without significant difference in all groups (**Fig. 8e**). Combination or AXL-CAR T treatment induced downregulation of CD206⁺ expression from CD68⁺ macrophages in tumours (**Fig. 8f**), which is consistent with decreased M2 polarization. In contrast, MWA did not induce downregulation of CD206⁺ expression of macrophages (**Fig. 8f**). This may be due to AXL inhibition of macrophages by AXL-CAR T cells. The similar observations were reported that TAM receptor inhibition decreased M2 characteristics of tumour-associated macrophages (**ref 33/34**).

We further evaluated the safety of AXL-CAR T cells in huNOG-EXL mice. There was no significant difference among all groups in terms of leukocyte percentages both in tumour and spleen, including T (CD3⁺), B (CD19⁺), NK (CD56⁺) and monocyte cells (CD14⁺) (**Fig. 8g/h**). Moreover, no obvious damage was observed in the organs from humanized NOG-EXL mice treated with MWA and/or AXL-CAR T cells by HE

staining (Fig. 8i). Together, these results indicated that combination treatment may be effective with low toxicities in humanized immunocompetent mice.

Fig. 8. Antitumour efficacy and safety of MWA +/- AXL-CAR T cells against A549-derived tumour in humanized immunocompetent mice. a. Schematic representation of a combination of MWA and AXL-CAR T cell administration. Briefly, humanized NOG-EXL immunocompetent mice received a subcutaneous injection of 5×10^6 A549 cells in the flank. When tumour volume reached about 200 mm³, mock, MWA +/- AXL-CAR T cells (i.v.) were administrated against A549-bearing mice. b. Tumour volume was measured every three days. c/d. The percentages of total CD3⁺ T cells (c) and CAR T cells (GFP+) (d) and e/f. The percentages of total macrophages

(e) and M2 macrophages (f) by flow cytometry. g/h. The leukocyte percentage in peripheral blood (g) and spleen (h) was detected by flow cytometry. i. Histopathological analysis of murine organ tissues by haematoxylin and eosin (HE) staining (magnification $\times 200$). Scale bar represents 50 μm . Data summarize one independent experiment (a-i) and presented as the mean \pm SD (b-l). *P < 0.05, **P < 0.01, ***P < 0.001.

Other Comments:

The authors should comment on the concordance of their data regarding AXL expression in patient samples with previous publications (e.g. Linger et al, Oncogene, 2013)

Response: Thanks for your suggestion. We discussed the AXL expression in patient samples with previous publications (**line 15-16, page 28, ref 23**).

Are all of the EGFR-mt samples in Table S2 from EGFR TKI-resistant tumors? If so, the table should be labeled as such to avoid confusion.

Response: Thanks for your mention. We labeled this information in **Table S2 (line 1, page 64)**.

In the results section, the figure panel reference for HLA-DR staining should be Figure 3f instead of Figure 3g

Response: Thanks for your mention. We modified the figure panel reference from Figure 3g to **Figure 3f (line 5, page 12)**.

Addition of the article “the” in this revised version is often unnecessary and detracts from clarity.

Response: Thanks for this critique. We deleted some “the” in the revised manuscript.

Reviewer 2 point 6 –The authors respond by stating that they have generated data to address the reviewers comment, but don’t show the data. The data should be included in the response to allow the reviewers to evaluate whether the concern has been

adequately addressed.

Response: Thanks for the comment.

The question of previous Reviewer 2 point 6:

-in light of their claim that MWA therapy could potentiate CAR-T effects. It would be keen for the authors to address the question whether MWA could help reduce the dose of CAR-T cells infused, thus limiting potential side effects and toxicity.

Response to Reviewer 2 point 6: In our previous preliminary study, we noticed that in A549-bearing NSG mice, combination group (MWA+ 5×10^6 AXL-CAR T cells) exhibited better antitumour efficacy than AXL-CAR T cell group (10^7 AXL-CAR T cells) alone, despite the dosage of CAR T cells were twice higher in the AXL-CAR T cell group. Therefore, we deduced that MWA may reduce the dose of CAR-T cells infused (see figure below).

Figure legend. Combination treatment showed better antitumour efficacy and reduced dosage of CAR T cells. NSG mice received a subcutaneous injection of 2×10^6 A549 cells. When tumour volume reached about 200 mm^3 , mock, AXL-CAR T (10^7 , i.v.) or MWA+AXL-CAR T cells (5×10^6 , i.v.) were administrated. Tumour volume was measured every three days.

P11 line 23: clarify that HCC827 is an Axl-negative cell line

Response: Thanks for your advice. We added this information accordingly (line 23, page 11).

Need to cite references describing clinical development of AXL antibodies to support the following statement “Moreover, a high number of AXL-targeted monoclonal antibodies, such as hMAb173 for renal cell carcinoma, D9 and E8 for pancreatic

cancer23, and YW327.6S2 for NSCLC and breast cancer, have shown anticancer efficacy in preclinical and earlier clinical trial stages.”

Response: Thanks for your mention. We cited references for this sentence (line 17-19, page 5, ref 16-17, 22-24).

Reviewer #6 (Remarks to the Author): to replace Reviewer #4

This is a very interesting study, with many innovations. But there are a few questions the author needs to answer.

1. What are the noteworthy results?

(1). The data in Fig. 3 are insufficient to show that MWA promotes antitumour activity of AXL-CAR T cells against lung cancer. In this part of the experiment, subcutaneous tumor models were used and 10 W and 40s as parameters sufficient to achieve partial ablation, so the tumor reduction seen in Fig. 3 b/d and Fig. 3 c/e may be caused by physical thermal ablation of MWA rather than enhancing the antitumor effect of AXL-CAR T. Fig. 3 g showed that there was no difference in the number of metastases between AXL-CAR T and MWA + AXL-CAR T groups, and Fig. 3 h showed no difference in lung weight, which also did not indicate that MWA enhanced the anti-tumor effect of AXL-CAR T. Therefore, is there more data to support the conclusion that MWA promotes antitumour activity of AXL-CAR T cells against lung cancer?

Response: We appreciate the professional comments.

It is true that in combination group in Fig. 3b/d and Fig. 3c/e, MWA resulted in partial ablation and reduced tumour burden. However, neither MWA nor AXL-CAR T cell monotherapy suppressed the tumour growth at the end of the study. In contrast, combination treatment yielded best antitumour effect among the four groups, these data suggested that combination strategy may enhance antitumour activity of AXL-CAR T cells against lung cancer.

Moreover, though Fig. 3g showed that there was no difference in the number of metastases between AXL-CAR T and MWA + AXL-CAR T groups when analysed by one-way ANOVA, there are significant difference when analysed by Student's t test (*P*

< 0.05). Thus, MWA enhances the antitumour effect of AXL-CAR T cells in terms of metastasis inhibition.

In the continued PDX study in **Fig. 4/5**, we found that MWA remodelled the TME in NSCLC PDX tumours by reducing HA, increasing vascularisation, decreasing IFP, and alleviating hypoxia. These favourable changes induced better tumour vascularisation, permeability and T cell infiltration, these changes may be beneficial for CAR T cells to perform their function and eventually resulted in better antitumour efficacy in combination group.

In addition, compared with mono-CAR T cells, our data indicated that the comb-CAR T cells displayed higher cytokines release, more activation/memory differentiation and less exhaustion markers than mono-CAR T cells (**Fig. 6**). This further indicated that MWA may enhance CAR T cell function and persistence.

Taken together, these data are sufficient to support the conclusion that MWA promotes antitumour activity of AXL-CAR T cells against lung cancer.

(2). Fig7 i Transmission electron microscopy results came from day42 tumor tissue, whether it is the result after preparing a single cell suspension for tumor tissue? Because the background is too clean, unlike the result of directly making ultrathin sections of tumor tissue.

Response: Thanks for your question. The used cells in Fig. 7i were sorted from tumour-infiltrating CAR T cells.

2. Will the work be of significance to the field and related fields? How does it compare to the established literature? If the work is not original, please provide relevant references.

Lung cancer has the highest mortality rate worldwide, with NSCLC accounting for about 85% and 5-year survival rate of only 5%. Improving the treatment status and achieving long-term survival is the most urgent need for patients with advanced NSCLC. In this work, AXL-CAR T combined with MWA was constructed to effectively inhibit the progression of NSCLC, providing an innovative theoretical

basis for the treatment of NSCLC. Compared with previous studies, first, this work investigated the role of AXL-CAR T in NSCLC for the first time, and cleverly combined MWA technology to enhance the anti-tumor effect of AXL-CAR T; second, this work also demonstrated that MWA reprogrammed the metabolic pattern of AXL-CAR T while the latter obtained a more activated and memory phenotype; and finally, it was unexpected that this work found that MWA reduced the level of hyaluronic acid, increased higher blood flow rate, elevated partial pressure of O₂, and then remodeled the tumor microenvironment. In summary, the work will be of significance to the field and related fields.

Response: We really appreciate your kind comments.

3. Does the work support the conclusions and claims, or is additional evidence needed?

This work requires additional work demonstrating that MWA promotes antitumour activity of AXL-CAR T cells against lung cancer.

Response: Thanks for your comments. As we responded in your **question 1**, we consider that current data in our manuscript are sufficient to demonstrate that MWA promotes antitumour activity of AXL-CAR T cells against lung cancer.

4. Are there any flaws in the data analysis, interpretation and conclusions? Do these prohibit publication or require revision?

The data analysis, interpretation and conclusions of this work are not flawed, except for the conclusion that " MWA promotes antitumour activity of AXL-CAR T cells against lung cancer " which requires more data support.

Response: Thanks for your comments. As we answered in your **question 1 and 3**, it is reasonable to make the above conclusion if take all the data into consideration.

5. Is the methodology sound? Does the work meet the expected standards in your field?

The methodology in this work is sound. The work meets the expected criteria in lung cancer and MWA field.

Response: We appreciate for your kind comments.

6. Is there enough detail provided in the methods for the work to be reproduced?

There is enough detail provided in the methods for the work to be reproduced.

Response: We appreciate for the kind comments.

REVIEWERS' COMMENTS

Reviewer #5 (Remarks to the Author):

The authors have added several studies in murine models and a more comprehensive discussion of their findings that enhance the impact of the manuscript. These additions satisfy my concerns.

RESPONSE TO REVIEWER COMMENTS

Reviewer #5 (Remarks to the Author):

The authors have added several studies in murine models and a more comprehensive discussion of their findings that enhance the impact of the manuscript. These additions satisfy my concerns.

Response: We appreciate for your kind comments.